# Transactive Response DNA-Binding Protein (TARDBP/TDP-43) Regulates Cell Permissivity to HIV-1 Infection by Acting on HDAC6

**DOI:** 10.3390/ijms23116180

**Published:** 2022-05-31

**Authors:** Romina Cabrera-Rodríguez, Silvia Pérez-Yanes, Rafaela Montelongo, José M. Lorenzo-Salazar, Judith Estévez-Herrera, Jonay García-Luis, Antonio Íñigo-Campos, Luis A. Rubio-Rodríguez, Adrián Muñoz-Barrera, Rodrigo Trujillo-González, Roberto Dorta-Guerra, Concha Casado, María Pernas, Julià Blanco, Carlos Flores, Agustín Valenzuela-Fernández

**Affiliations:** 1Laboratorio de Inmunología Celular y Viral, Unidad de Farmacología, Sección de Medicina, Facultad de Medicina, Universidad de La Laguna (ULL), 38320 La Laguna, Spain; rcabrerr@ull.edu.es (R.C.-R.); sperezya@ull.edu.es (S.P.-Y.); jesteveh@ull.edu.es (J.E.-H.); jgarcial@ull.edu.es (J.G.-L.); rotrujil@ull.edu.es (R.T.-G.); 2Genomics Division, Instituto Tecnológico y de Energías Renovables, 38600 Santa Cruz de Tenerife, Spain; rgonzalezmontelongo@iter.es (R.M.); jlorenzo@iter.es (J.M.L.-S.); ainigo@iter.es (A.Í.-C.); lrubio@iter.es (L.A.R.-R.); amunoz@iter.es (A.M.-B.); cflores@ull.edu.es (C.F.); 3Analysis Department, Faculty of Mathematics, Universidad de La Laguna (ULL), 38296 La Laguna, Spain; 4Mathematics, Statistics and Operations Research, Faculty of Mathematics, Universidad de La Laguna (ULL), 38320 La Laguna, Spain; rodorta@ull.edu.es; 5Unidad de Virología Molecular, LRIR, Centro Nacional de Microbiología (CNM), Instituto de Salud Carlos III, 28029 Madrid, Spain; ccasado@isciii.es (C.C.); mpernas@isciii.es (M.P.); 6AIDS Research Institute IrsiCaixa, Institut de Recerca en Ciències de la Salut Germans Trias i Pujol (IGTP), 08916 Badalona, Spain; jblanco@irsicaixa.es; 7Faculty of Medicine, University of Vic-Central University of Catalonia (UVic-UCC), 08500 Barcelona, Spain; 8Research Unit, Hospital Universitario N. S. de Candelaria, 38010 Santa Cruz de Tenerife, Spain; 9CIBER de Enfermedades Respiratorias, Instituto de Salud Carlos III, 28029 Madrid, Spain; 10Instituto de Tecnologías Biomédicas (ITB), Universidad de La Laguna (ULL), 38200 San Cristóbal de La Laguna, Spain

**Keywords:** TARDBP/TDP-43, HDAC6, pore fusion formation, cell-permissivity, HIV-1 infection

## Abstract

The transactive response DNA-binding protein (TARDBP/TDP-43) influences the processing of diverse transcripts, including that of histone deacetylase 6 (HDAC6). Here, we assessed TDP-43 activity in terms of regulating CD4+ T-cell permissivity to HIV-1 infection. We observed that overexpression of wt-TDP-43 increased both mRNA and protein levels of HDAC6, resulting in impaired HIV-1 infection independently of the viral envelope glycoprotein complex (Env) tropism. Consistently, using an HIV-1 Env-mediated cell-to-cell fusion model, the overexpression of TDP-43 levels negatively affected viral Env fusion capacity. Silencing of endogenous TDP-43 significantly decreased HDAC6 levels and increased the fusogenic and infection activities of the HIV-1 Env. Using pseudovirus bearing primary viral Envs from HIV-1 individuals, overexpression of wt-TDP-43 strongly reduced the infection activity of Envs from viremic non-progressors (VNP) and rapid progressors (RP) patients down to the levels of the inefficient HIV-1 Envs observed in long-term non-progressor elite controllers (LTNP-EC). On the contrary, silencing endogenous TDP-43 significantly favored the infectivity of primary Envs from VNP and RP individuals, and notably increased the infection of those from LTNP-EC. Taken together, our results indicate that TDP-43 shapes cell permissivity to HIV-1 infection, affecting viral Env fusion and infection capacities by altering the HDAC6 levels and associated tubulin-deacetylase anti-HIV-1 activity.

## 1. Introduction

Transactive response DNA-binding protein (TARDBP), also known as transactive response DNA binding protein 43 kDa (TDP-43), is a nuclear RNA-binding protein that is able to process RNA, thereby acting in different biological processes, including splicing, transcription and translation, mRNA transport, mRNA stability and pri-miRNA processing [1,2,3,4,5,6,7,8,9]. Since TDP-43 recognizes UG-rich sequences present within approximately one-third of all transcribed genes [5,6,10], it is uniquely able to influence the processing of hundreds to thousands of transcripts. One of the transcripts that is mainly regulated by TDP-43 is that corresponding to the cytoplasmic enzyme, histone deacetylase 6 (HDAC6) [11]. TDP-43 binds specifically and functionally to HDAC6 mRNA. A TDP-43 knockout of Drosophila melanogaster has validated its biological relevance, further confirmed by the specific downregulation of HDAC6 [11]. Hence, loss of functional TDP-43 causes HDAC6 downregulation and might thereby contribute to several pathogenic biological events where these two factors could be functionally involved [12,13,14,15,16,17,18,19]. Interestingly, TDP-43 contains two RNA-recognition motifs (RRMs) and a C-terminal glycine-rich domain (GRD) [20,21,22]. TDP-43 was originally reported as a novel protein binding to transactivator-responsive DNA sequences within human immunodeficiency virus type 1 (HIV-1), acting as a strong transcriptional repressor [20]. However, it has been reported that TDP-43 has no effect on viral LTR transcription in cells [23]. Thus, it did not affect HIV-1 gene expression and virus production in T cells and macrophages [23].

It is important to note that HDAC6 is key to regulating HIV-1 infection. HDAC6 affects pore fusion formation by preventing HIV-1-mediated α-tubulin acetylation of MTs, and promoting HDAC6/p62 autophagy to clear key viral factors, such as Vif and Pr55Gag, thereby stabilizing the HDAC6/APOBEC3G (A3G) restriction factor complex [24,25,26,27,28]. Furthermore, there is a clear-cut correlation between the inability of primary HIV-1 envelope complex (Env) of virus from elite controller individuals (ECs) to signal and overcome the HDAC6 barrier, and this HIV-1 long-term non-progressor phenotype [29]. On the contrary, functional primary Envs from virus of viremic HIV-1 patients (Progressors, Rapid Progressors (RPs) and Viremic non-progressors (VNPs)) are able to overcome the HDAC6 barrier [26,29]. Therefore, it is plausible that a cellular factor that can balance cellular HDAC6 levels could regulate its anti-HIV functions and cell permissivity to HIV-1 infection.

Altogether, there is a lack of evidence involving TDP-43 in the regulation of HIV-1 viral entry and infection. The authors propose that due to the importance of HDAC6 for limiting HIV-1 fusion and infection in the first step of the viral cycle, TDP-43 conditions cell permissivity to fuse and become infected by HIV-1 by regulating HDAC6 expression at mRNA and protein levels. Our results indicate that the TDP-43/HDAC6 axis could be crucial for regulating HIV-1 infection.

## 2. Results

### 2.1. Characterization of the Expression and Cellular Localization of TDP-43

To study the role of TDP-43 in the control of HIV-1 infection, we first analyzed the pattern of expression of TDP-43 in permissive CEM.NKR-CCR5 T cells. In non-transfected cells, we observed that endogenous HDAC6 predominantly distributed at the cytoplasm, being also located at the nucleus (Figure 1A, quantified in Figure 1B,C), whereas endogenous TDP-43 mainly localizes at the nucleus (Figure 1A, quantified in Figure 1B,C). We observed that transiently overexpressed Flag-wt-TDP-43 (N-terminal Flag-tagged, wild-type (wt) TDP-43 construct) mainly localizes in the isolated nuclear fraction of cells as observed by biochemical detection (Figure 1, Flag-associated Western-blot bands), being also detected in the cytoplasmic fraction, similarly as in the case of endogenous TDP-43 (Figure 1, lower molecular weight TDP-43 Western-blot bands). As a control, we used a N-terminal Flag-tagged NLS-TDP-43 mutant (NLS-mut-TDP-43) that lacks the nuclear localization signal (NLS) of the protein [11,30,31,32].

Moreover, it has been reported that this cytoplasmic ΔNLS TDP-43 mutant perturbs the trafficking of the endogenous TDP-43 between the nucleus and the cytoplasm [30]. The results obtained indicate that this NLS-TDP-43 mutant is located at the cytoplasm, as observed by biochemical detection of cell fractioning (Figure 1A, *Flag and TDP-43 associated Western-blot bands*) and confocal microscopy (Figure 1C, *Flag-mut-TDP-43 series of images*). Moreover, we observed that in the presence of the Flag-NLS-mutant-TDP-43 construct, the endogenous TDP-43 was slightly detected at the nucleus (Figure 1A, *track 6*), being accumulated at the cytoplasm together with the mutant construct (Figure 1A, *track 3*). Thus, nuclear endogenous TDP-43 (Figure 1A, *detected in control track 4 (associated cytoplasmic control track 1)* and Flag-wt-TDP-43 condition *(track 5 and associated cytoplasmic track 2)*) is not detected at the nuclear fraction of the Flag-NLS-mut-TDP-43 experimental condition (Figure 1A, *track 6*). This fact was previously reported [30]. Furthermore, we observed that the ΔNLS TDP-43 mutant abnormally co-localizes with endogenous HDAC6 (Figure 1C, *Flag-mut-TDP-43 Merge images*). This fact is not observed with the endogenous TDP-43 or the wt-TDP-43 construct (Figure 1C, *endogenous and Flag-wtTDP-43 series of images*). Since it has been reported that the NLS mutant affects the localization and function of the endogenous TDP-43 [11,30,31,32], and considering that we observed an abnormal pattern of distribution with HDAC6 that differs from that observed with full-length TDP-43, we will avoid working with the Flag-NLS-mutant-TDP-43 construct.

The quantification of the level of expression of these two TDP-43 constructs is shown in Figure 1B. Similar results are obtained when these cells are analyzed by fluorescent confocal microscopy (Figure 1C). Therefore, endogenous TDP-43 and Flag-wt-TDP-43 are both distributed in the nucleus (Figure 1C, *see associated series of images*) and do not co-distribute with HDAC6.

### 2.2. TDP-43 Stabilizes HDAC6 and Diminishes α-Tubulin Acetylation

We observed that in these HIV-1 permissive CEM.NKR-CCR5 T cells the overexpression of TDP-43 constructs enhance the protein level of HDAC6 (Figure 1A,B, *Western-blot bands and associated histograms, respectively*). It is thought that TDP-43 stabilizes HDAC6 cells levels by acting on the mature HDAC6 mRNA [11]. In this matter, we have observed the stabilization of higher levels of HDAC6 mRNA in cells overexpressing the TDP-43 constructs compared to control cells, by quantification of isolated and specifically sequenced HDAC6 mRNA (Figure 1D,E). By these techniques we determined that we have overexpressed similar levels of the two TDP-43 constructs (Figure 1D,E), as confirmed by biochemical detection of the associated proteins (Figure 1A,B).

The increase in the level of the HDAC6 tubulin-deacetylase enzyme, by overexpression of wt-TDP-43, induces a diminution in the level of acetylation of α-tubulin subunit of the microtubules (MTs) (Figure 1A, *acetylated*
*α-tubulin in the cytoplasmic fraction*), a post-translational modification associated with stable MTs [33,34,35,36,37,38,39]. This effect is not observed by the overexpression of the NLS-mut-TDP-43 construct (Figure 1A). Perhaps its abnormal cytoplasmic localization in aggregates where HDAC6 co-localizes may alter the HDAC6 tubulin-deacetylase activity, as reported in relation to the interaction of TDP-43 with p62 and HDAC6 at the cytoplasm [15,40,41,42,43,44]. Here, we used this mutant construct as a control for studying TDP-43 cellular localization. We observed that overexpression of the wt-TDP-43 construct stabilizes mRNA and protein levels of the HDAC6 enzyme, which in turn promotes the deacetylation of acetylated α-tubulin, as we describe here. Therefore, we searched for the TDP-43 function on regulating cell permissivity to HIV-1 infection by working with wt-TDP-43 construct or by specific siRNA-TDP-43 silencing.

### 2.3. TDP-43 Overexpression Inhibits HIV-1 Entry and Infection

First, we studied the effect of wt-TDP-43 on the early infection of permissive CEM.NKR-CCR5 T cells with HIV-1. As shown in Figure 1, cells overexpressing wt-TDP-43 were infected by using HIV-1, luciferase-reporter pseudovirus bearing a reference CCR5 tropic (R5-tropic) BaL envelope (Env) complex or an X4-tropic HXB2 Env (a reference CXCR4 tropic Env). We observed that overexpression of wt-TDP-43 did not significantly change the cell-surface expression levels of CD4, CCR5 and CXCR4 molecules (i.e., main receptor and co-receptors for HIV-1 infection, respectively) (Figure 2A). The cell infection using pseudovirus bearing BaL or HXB2 Env was notably diminished in cells overexpressing wt-TDP-43 compared to control, untreated cells (Figure 2B). Therefore, the reduction in the permissivity to HIV-1 infection of these CD4+ T cells appears to be independent of the viral tropism. The TDP-43-mediated increase in the level of the HDAC6 enzyme and the subsequent diminution of the levels of acetylated α-tubulin in MTs (Figure 1) promote a cellular environment that limits early HIV-1 infection, as previously reported [24,26,29]. Hence, overexpression of wt-TDP-43 negatively regulates early viral entry and infection, an antiviral activity that at least accounts for the associated increase in the HDAC6-tubulind deacetylase antiviral action.

### 2.4. TDP-43 Overexpression Inhibits HIV-1 Env-Mediated Pore Fusion Formation

As reported, HDAC6 tubulin-deacetylase activity limits HIV-1 Env-mediated pore fusion. HDAC6 impairs HIV-1-mediated stabilization of a cortex of stable and acetylated MTs in their α-tubulin subunits, which is required for efficient viral entry and infection, and is directly dependent on the first viral Env/CD4 interaction and signaling [24,26,29]. We then studied the effect of wt-TDP-3 in the control of HIV-1 Env-mediated pore fusion formation in an Env-mediated cell-to-cell fusion model [24,28,45,46,47] (see Materials and Methods). In Flag-wt-TDP-43 transfected HeLa P5 cells, we observed an increase in HDAC6 with a concomitant decrease in the levels of acetylated MTs, as compared to control cells (Figure 3A, *total cell lysate, quantified in the associated histograms below*).

Moreover, the biochemical analysis of protein expression and localization in cellular fractions indicates that endogenous HDAC6 is predominantly distributed at the cytoplasm, and is also located in the nucleus, whereas endogenous TDP-43 mainly localizes in the nucleus (Figure 3A, *cell fractioning, quantified in the associated histograms below*). We observed that transiently overexpressed Flag-wt-TDP-43 construct mainly localizes in the isolated nuclear fraction of cells (Figure 3A, *cell fractioning, quantified in the associated histograms below*). Under this experimental condition, we observed an increase in the amount of HDAC6 enzyme with a concomitant decrease in acetylated MTs (Figure 3A, *cell fractioning, HDAC6 and acetylated*
*α-tubulin in the associated cytoplasmic fractions, quantified in the associated histograms below*).

Therefore, the overexpression of wt-TDP-43 (Figure 3A) did not negatively affect the cell-surface expression of CD4, CCR5 and CXCR4 on HeLa P5 cells (Figure 3B), increasing the amount of the HDAC6 protein, subsequently diminishing the level of acetylated MTs (Figure 3A). Furthermore, we have observed the higher stabilization levels of HDAC6 mRNA in cells overexpressing the wt-TDP-43 construct compared to control cells, by quantification of isolated HDAC6 mRNA (Figure 3C). Under these experimental conditions, our results indicate that the co-culture of these HeLa P5 with HeLa cells expressing 243 Env (X4-tropic) or ADA Env (R5-tropic) led to a reduction of HIV-1 Env-mediated cell-to-cell fusion, being a direct measurement of the inhibition of HIV-1 Env-mediated pore fusion formation in cells overexpressing wt-TDP-43 (Figure 3D). Altogether, these results indicate that the efficiency of HIV-1 infection and Env-mediated cell fusion is inversely related to HDAC6 activity which is under the control of TDP-43.

### 2.5. Specific siRNA Silencing of Endogenous TDP-43 Decreases HDAC6 Protein Level, Increases α-Tubulin Acetylation and Enhances Cell Permissivity to HIV-1 Infection

We screened four different previously reported [11] siRNA-TDP-43 oligos (i.e., siRNA-TDP-43 A to D; Figure 4A) to assay their ability to specifically interfere TDP-43 mRNA and knockdown the protein in permissive CEM.NKR-CCR5 T cells. Three of them present good activity to specifically silence TDP-43 (Figure 4A, *siRNA-TDP-43 B, C and D oligos-associated Western-blot bands*, and Figure 4B, *quantified in associated histogram*). Moreover, we quantified the amount of the mRNA of TDP-43 and HDAC6 present in cells nucleofected with the four siRNA-TDP-43 oligos and some combination of them (Figure 4C), by direct quantification by sequencing of their mRNAs, and also by RT-qPCR (see Materials and Methods) (Figure 4C,D, respectively). The silencing of TDP-43 decreases the level of the HDAC6 enzyme with subsequent stabilization of the levels of acetylated α-tubulin (Figure 4A, *HDAC6 and acetylated α-tubulin-associated Western-blot bands, and*
Figure 4B, *quantified in histograms*), thereby promoting an appropriate cellular environment for HIV-1 viral infection.

Thus, we infected TDP-43-silenced cells with synchronous doses of HIV-1 (luciferase-reporter pseudovirus) bearing the R5-tropic BaL Env or the X4-tropic HXB2 Env. We observed that specific siRNA silencing of endogenous TDP-43 did not negatively affect the cell-surface expression of CD4, CCR5 and CXCR4 (Figure 5A). CEM.NKR-CCR5 T cells were easier to infect when TDP-43 was silenced, expressing low levels of HDAC6 and increased levels of acetylated α-tubulin in MTs, when compared to control cells (Figure 5B). Once again, the effect observed on HIV-1 infection was independent of the viral tropism (Figure 5B).

We then used a combination of two oligos, siRNA-TDP-43 B and C, in order to silence TDP-43. Under this experimental condition, we assayed HIV-1 Env-mediated pore fusion formation, entry and infection. We first analyzed how siRNA-TDP-43 silencing affected the ability of the viral Env to promote pore fusion formation, by using the HIV-1 Env-mediated cell-to-cell fusion model. TDP-43-silenced HeLa P5 cells were co-cultured with either HeLa 243 (X4-tropic Env) or HeLa ADA (R5-tropic Env) cells, observing an increase in the Env-mediated cell-to-cell fusion activity when compared to scrambled control cells (Figure 6). This positive effect on HIV-1 Env-mediated cell-to-cell fusion correlated with low levels of the anti-fusogenic enzyme HDAC6, with a concomitant increase in acetylated MTs in α-tubulin in cells where TDP-43 has been knocked-down. The enhancement of the fusogenic activity of HIV-1 Env after TDP-43 knockdown was not related to the viral Env tropism (Figure 6).

Therefore, the degree of permissivity to HIV-1 infection of these CD4+ T cells appears to be dependent on the level of TDP-43 which conditions the expression level of the antiviral HDAC6 enzyme and MT acetylation in α-tubulin.

### 2.6. TDP-43 Affects the Viral Function of Primary Viral Envs from Virus of HIV-1 Individuals with Different Clinical Phenotypes

We wanted to seek the effect of TDP-43 on cell permissivity against primary Env from HIV-1 individuals with different clinical phenotypes, such as long-term non-progressors, elite controllers (LTNP-ECs), viremic non-progressors (VNPs) and rapid progressors (RPs) that we have characterized before in their viral function [26,29]. We observed that HIV-1 Envs from virus of LTNP-EC individuals that control viral infection present less infection activity compared to viral Envs of viremic HIV-1 individuals (VNPs) and to those that do not control viral infection at all (RPs) (Figure 7), as previously reported [26,29]. Irrespective of the clinical phenotype, in all functional Envs, overexpression of wt-TDP-43 impaired HIV-1 Env-mediated infection (Figure 7A), revealing that VNP and RP Envs lost their functionality, achieving the levels observed with the inefficient LTNP-EC Envs.

On the contrary, specific siRNA-TDP-43 (B + C oligos) knock-down enhanced the infection activity of the HIV-1 Env of VNP and RP individuals (Figure 7B). Remarkably, TDP-43 silencing renders cells more permissive even to inefficient LTNP-EC Envs (Figure 7B).

Altogether, these results prompted the suggestion that TDP-43 by conditioning HDAC6 mRNA and subsequent protein levels alters the permissive status of target cells against HIV-1 fusion and infection.

## 3. Discussion

In this work, we have observed that overexpression of wt-TDP-43, a cell factor that mainly localizes at the nucleus, stabilized the antiviral enzyme HDAC6, increasing its mRNA and protein levels. Under these experimental conditions, target cells were not efficiently infected by HIV-1 compared to untreated control cells. This TDP-43-mediated antiviral effect occurred with HIV-1 virus bearing a R5-tropic viral Env as well as an X4-tropic Env. The enhanced levels of the antiviral HDAC6 enzyme in cells overexpressing TDP-43 could be related to the anti-HIV-1 effect exerted by TDP-43. To confirm this TDP-43/HDAC6 antiviral relationship, we further analyzed the ability of HIV-1 Env to promote pore fusion formation in a well-characterized cell-to-cell fusion assay using target cells overexpressing wt-TDP-43. These treated cells increased the protein levels of the anti-fusogenic HDAC6 enzyme and when co-cultured with effector cells, stably expressing HIV-1 viral Env at cell-surface, cell-to-cell fusion occurred to a greater extent compared to the same experiment performed with control target cells. It has been reported that the level of expression of tubulin-deacetylase HDAC6 in target cells conditions HIV-1 Env-mediated pore fusion formation and early infection activities [24,29]. Thus, it is conceivable that TDP-43 negatively regulates this viral Env function by stabilizing HDAC6 mRNA and increasing protein levels, a process independent of the viral Env tropism. The specific siRNA silencing of endogenous TDP-43 in target cells consistently leads to an increase in the fusogenic and infection activities of the HIV-1 Env. In target cells where endogenous TDP-43 was silenced by using specific siRNA oligos, a significant decrease in the levels of HDAC6 occurred which favors HIV-1 Env-mediated fusion and infection.

These data indicate that TDP-43 regulates the level of the HDAC6 enzyme due to its stabilizing effect on HDAC6 mRNA, as previously reported [11], rather than interacting with the HDAC6 protein, since the two proteins do not co-distribute. The level of HDAC6 therefore conditions the amount of acetylated MTs that regulate HIV-1 Env-mediated pore fusion formation and infection [24,28]. In this regard, we have previously reported that α-tubulin acetylation is a key Env/CD4-mediated signal for productive HIV-1 fusion and infection [24,26,28,29]. Moreover, we have reported that tubulin-deacetylase HDAC6 negatively controls, either the endogenous activity of the enzyme or the overexpressed functional deacetylase, HIV-1-mediated pore fusion formation and infection by lowering the levels of acetylated MTs [24]. It means that the level of expression of HDAC6 and the acetylation status/level of MTs determine the capacity of HIV-1-Env to promote membrane fusion (pore fusion formation), entry and infection [24,26,28,29]. Therefore, our results indicate that low levels of TDP-43 enhance target cell permissivity to HIV infection at least by negatively altering the cellular levels of the antiviral HDAC6 enzyme and increasing MT acetylation.

These observations were further confirmed by using primary Envs from virus of HIV-1 individuals with different clinical phenotypes. An increase in the level of expression of wt-TDP-43 strongly diminished the infection activity of HIV-1 Envs from virus of VNP and RP HIV-1 individuals, even reaching the levels of the inefficient HIV-1 Env of virus of LTNP-EC individuals. On the contrary, low levels of endogenous TDP-43, obtained after siRNA knocking-down, significantly favored the infection activity of primary HIV-1 Envs of VNP, progressors and HIV-1/LTNP-EC individuals. The later observation is particularly interesting, since it provides a clear demonstration that defective viral features observed in these LTNP-EC individuals [27,29,48] are perhaps also modulated by these two cellular factors. These data clearly show that TDP-43 regulates cell permissivity to HIV-1 infection, independently of the viral Env tropism and strain, by affecting viral Env fusion and infection capacities, thereby altering the cellular level of the HDAC6 antiviral factor (summarized in the schemes of Figure 8). Although it was observed that HDAC6 mRNA is targeted by TDP-43, thereby affecting HDAC6-mediated anti-HIV-1 activity, this might not be the only mRNA associated with proteins that may affect HIV-1 infection targeted by TDP-43.

The TDP-43/HDAC6 axis therefore regulates cell permissivity to HIV-1 infection. This event may have negative consequences in HIV-1 LTNP-EC individuals, particularly if a negative regulation of TDP-43 occurs with a concomitant decrease in HDAC6 that renders cells more permissive against inefficient LTNP-EC Envs, thereby favoring viral infection. In this regard, it has been reported that the ability of the viral Env to trigger signals that overcome the HDAC6 barrier is directly related to its fusion and infection activities [24,29]. The TDP-43/HDAC6 axis could be another factor that is worth exploring as well as the immune responses in EC individuals that lose their status of natural controllers of the infection [29,49,50].

The authors have previously reported the anti-HIV-1 activity of HDAC6 by targeting the Pr55Gag and Vif key viral proteins to the autophagy degradative pathway, inhibiting viral production and virion infectiveness but promoting the stabilization of the anti-HIV-1 restriction factor A3G [25,51]. Furthermore, productive early HIV-1 infection requires the inhibition of autophagy and its related machinery, such as the pro-autophagic and anti-HIV-1 HDAC6 enzyme, whereas non-productive signaling in bystander target cells promotes late autophagy and subsequent cell death [25,26,29,52,53,54]. Therefore, it is plausible that the TDP-43 action on HDAC6 could also control HIV-1 infection and replication by modulating the HDAC6-mediated autophagy anti-HIV-1 functions.

Regarding TDP-43 and other viral infections, it has been reported that in cells infected by coxsackievirus B3 (CVB3), TDP-43 is translocated from the nucleus to the cytoplasm through the activity of the 2A viral protease. This viral protease cleaves TDP-43, affecting its solubility and promoting protein aggregates [55]. The present work shows that knockdown of TDP-43 results in an increase in viral titers, suggesting a protective role for TDP-43 in CVB3 infection [55]. Studies of the human endogenous retrovirus-K (ERV) indicates that functional TDP-43 fails to transcriptionally regulate ERVK expression [56], similarly as reported in cells for the activation of the HIV-1 LTR [23]. However, mutant forms of TDP-43, associated with amyotrophic lateral sclerosis (ALS), promoted ERVK protein aggregation [56], a virus associated with motor neuron damage and ALS [56,57,58,59]. In this regard, human ERVK proteins accumulate in cortical neurons of patients with HIV-1 infection [60], where high ERVH-RT activity has been reported [61] and linked with the accumulation of pathogenic TDP-43 and associated HIV-1 neuropathology [61]. In the central nervous system (CNS), the imbalance between TDP-43 and HDAC6 levels of expression and functions seems to be related to neurodegeneration and disease [12,13,14,15,16,17,18,19]. In this matter, it has been reported that HDAC6 prevents the neurotoxicity of the HIV-1-Env gp120 subunit in cortical neurons [62]. Maybe the TDP-43/HDAC6 axis in the CNS could act to protect against HIV-1 toxicity in this tissue, whereas an imbalance in their expression levels and functions could favor HIV-1 damage, but this needs to be explored further.

Therefore, TDP-43 appears to regulate cell permissivity to HIV-1 infection by conditioning viral Env-mediated fusion and infection capacities by modulating the level of expression of the antiviral HDAC6 tubulin-deacetylase enzyme.

## 4. Materials and Methods

### 4.1. Antibodies and Reagents

Rabbit anti-HDAC6 (H-300; sc-11420) and polybrene (sc-134220) were obtained from Santa Cruz Biotechnology (Santa Cruz, CA, USA). The neutralizing monoclonal antibody (mAb) RPA-T4 (eBioscience, San Diego, CA, USA) directed against CD4 was phycoerythrin (PE)-labelled (for flow cytometry). PE conjugates of the CD184 (clone 12G5) and CD195 (clone 2D7/CCR5) (BD Bioscience/BD PharMingen, San Jose, CA, USA) are directed against the second extracellular loop of CXCR4 and CCR5, respectively. Secondary Alexa Fluor 568-goat anti-mouse (to label TDP-43) and Alexa Fluor 488-goat anti-rabbit (to label HDAC6) were purchased from Molecular Probes (Eugene, OR). Rabbit anti-TDP-43 (Catalog Number T1705); mouse anti-TDP-43 mAb (clone 2E2-D3, catalog Number WH0023435M1); mAb anti-Flag M2 (F1804); mAbs anti-α-tubulin (T6074) and anti-acetylated α-tubulin (T7451); and secondary horseradish peroxidase (HRP)-conjugated Abs, specific for any Ab species assayed, were purchased from Sigma-Aldrich (Sigma-Aldrich, St. Louis, MO, USA). Complete™ Protease Inhibitor Cocktail (11697498001) was obtained from Roche Diagnostics (GmbH, Mannheim, Germany).

### 4.2. DNA Plasmids and Viral DNA Constructs

Vectors for expression of wild-type TDP-43 (Flag-wt-TDP-43) or mutant at the import NLS signal (Flag-NLS-mut-TDP-43) were provided courtesy of Dr. Thorsten Schmidt [11]. The pNL4-3.Luc.R-E- provirus (∆*nef*/∆*env*), the R5-tropic BaL.01-envelope (*env*) glycoprotein plasmid (catalog numbers 6070013 and 11445, respectively), the X4-tropic HXB2-envelope (*env*, catalogue number 5040154), and the pHEF-VSV-G vector (catalogue number 4693) encoding the vesicular stomatitis virus G (VSV-G) protein (i.e., for virus production control) were obtained via the NIH AIDS Research and Reference Reagent Program (http://www.aidsreagent.org/, accessed on 4 May 2022).

### 4.3. Primary Env Clones and Ethics Statement

Full-length Env expression plasmids isolated from HIV-1 rapid progressors (RP; RP7C4 and RP7C12 Envs), long-term non-progressor elite controllers (LTNP-EC; LTNP1_12 and LTNP3_3 Envs) and viremic non-progressor (VNP; VNP9C10 and VNP11C13 Envs) individuals were obtained and characterized as previously described [26,29]. Samples were obtained from participants in previous studies who gave informed consent for this study. Consents were approved by the Ethical and Investigation Committees for their corresponding hospital and the samples were encoded and de-identified in these centers. All clinical investigations were conducted according to the principles expressed in the Declaration of Helsinki in 1975, as revised in 1983. For LTNP-EC Envs, the studies were approved by the “Comité de ética de la Investigación y de Bienestar Animal of the Instituto de Salud Carlos III” under no. CEI PI 05_2010-v3 and CEI PI 09-2013v2, and for VNP and RP Envs were approved by the Ethics Committee of the Hospital Germans Trias I Pujol (no. E0-12-017). All methods were carried out in accordance with relevant guidelines and regulations. All procedures were approved by the Ethics Committee of the Hospital Germans Trias I Pujol. All individuals provided their written informed consent.

### 4.4. Cells

The human CEM.NKR-CCR5 permissive cell line (catalog number 4376, NIH AIDS Research and Reference Reagent Program) is a reference cell-based assay for the evaluation of HIV-1 infection and neutralization by Abs [64], being suitable for the general measurement of HIV-1 infection and neutralization assays by Abs or potential new anti-HIV-1 drugs, as we have also reported [25,26,28,29,45,65,66,67]. CEM.NKR-CCR5 and HEK-293T cells (catalog number 103, NIH AIDS Research and Reference Reagent Program) were grown at 37 °C in a humidified atmosphere with 5% CO2 in RPMI 1640 medium (Lonza, Verviers, Belgium) in the case of the CEM.NKR-CCR5 cells, and in DMEM (Lonza, Verviers, Belgium) when HEK-293T cells were used, both media supplemented with 10% fetal calf serum (Lonza, Verviers, Belgium), 1% L-glutamine, and 1% penicillin-streptomycin antibiotics (Lonza, Verviers, Belgium) and mycoplasma free (Mycozap antibiotics, Lonza, Verviers, Belgium), and both cell lines were regularly split every 2–3 days. The HEK-293T cell line was cultured to 50–70% confluence in fresh supplemented medium 24 h before cell transfection with viral or human DNA constructs. The HeLa-P5 cells are constitutively expressing CXCR4 and stably transfected with human CD4 and CCR5 cDNAs and with an HIV-1 long terminal repeat-driven-β-galactosidase reporter gene [24,46,47], as well as HeLa-243 and HeLa-ADA cells, co-expressing the Tat and X4- and R5-tropic HIV-1-Env proteins, respectively, were provided by Dr. M. Alizon (Hôpital Cochin, Paris, France) and cultured as previously described [24,46,47]. The HEK-293T cell line was similarly cultured in supplemented DMEM (Lonza, Verviers, Belgium), as reported [25,26,29,45]. Cells were harvested and cultured to 50–70% confluence in fresh supplemented DMEM 24 h before cell transfection with viral or human DNA constructs.

### 4.5. Messenger RNA Silencing

We used the short interference RNA (siRNA) oligonucleotides (oligos) indicated below, specifically directed against the indicated mRNA sequences of TDP-43, in order to knockdown TDP-43 expression. Transient siRNA transfections were performed using Amaxa kits C (Amaxa Lonza, Verviers, Belgium) for CEM.NKR-CCR5 cells, or polyethylenimine when using HeLa P5, with 1 μM of a commercial scrambled control oligo or TDP-43-specific siRNAs: siRNA-TDP-43 A: 5′-CAAUAGCAAUAGACAGUUA[dT][dT]-3′; siRNA-TDP-43 B: 5′-CACUACAAUUGAUAUCAAA[dT][dT]-3′; siRNA-TDP-43 C: 5′-GAAUCAGGGUGGAUUUGGU[dT][dT]-3′; siRNA-TDP-43 D: 5′-GAAACAAUCAAGGUAGUAA[dT][dT]-3′ (Sigma-Aldrich, St. Louis, MO, USA) [11]. Cells nucleofected with scrambled or specific siRNA oligos against TDP-43 were lysed and analyzed with specific antibodies in Western blots to determine the endogenous TDP-43 silencing.

### 4.6. RNA Extraction and RT-qPCR

RNA from CEM.NKR-CCR5 cells was isolated using the RNeasy Mini kit (Qiagen, Hilden, Germany) following the manufacturer’s instructions. The cellular qRT–PCR, 1000 ng total RNA was reverse transcribed using iScript™ cDNA Synthesis Kit (Bio-Rad), random primers and anchored oligo-dT primer. RT reaction (20 μL) was used as template for transcript amplification; 1/10 dilutions were used in triplicate with 0.2 μM primer and 10 μL LightCycler 480 SYBR Green I Master in a 20 μL reaction and qPCR executed in a 98-well block on a CFX96 Touch Real-Time PCR Detection System (Bio-Rad, Hercules, CA, USA). Absolute transcript levels for TDP-43, HDAC6 and the controls were obtained by a second derivative method. Relative transcript levels were calculated as TDP-43 or HDAC6/controls ratio and normalized to the relative expression level of the mock-transfected control. The primers used for qPCR were as follows: for HDAC6 (Forward: 5′-ATGCAGCTTGCGGTTTTTGC-3′/Reverse: 5′-TGCTGAGTTCCATTACCGTGG-3′); for TDP-43 (Forward: 5′-GCTTCGCTACAGGAATCCAG-3′/Reverse: 5′-GATCTTTCTTGACCTGCACC-3′); and for GADPH as housekeeping gene (Forward: 5′-GGAAGCTCACTGGCATGGCCT-3′/Reverse: 5′-CGCCTGCTTCACCACCTTCTTG-3′).

### 4.7. RNA Sequencing

#### 4.7.1. Library Preparation and Sequencing

Libraries were prepared using 50 ng of total RNA (all with RIN > 8.5) according to the manufacturer’s instructions with the Illumina^®^ Stranded Total RNA Prep with Ribo-Zero Plus (Illumina, Inc., San Diego, CA, USA), including the depletion of cytoplasmic, mitochondrial and beta globin rRNA from 0.1–1 μg of intact total RNA samples. After rRNA depletion, libraries were prepared with 50 ng of total RNA (RIN > 8.5). Library integrity was confirmed by size analysis on a 4200 TapeStation (Agilent, Santa Clara, CA, USA) with a D1000 ScreenTape. All libraries were within an average size range of 360 to 400 bp. Library integrity and size (size range of 360 to 400 bp) was confirmed by analysis on a 4200 TapeStation (Agilent, Santa Clara, CA, USA) with D1000 ScreenTape. Library concentrations were determined using the dsDNA HS assay kit on Qubit 3.0 (Thermo Fisher Scientific, Waltham, MA, USA). Sequencing was performed with paired-end 75 cycles on a NextSeq 550 System (Illumina, Inc.) at the Instituto Tecnológico y de Energías Renovables.

#### 4.7.2. Bioinformatic Analysis

The raw sequencing data was demultiplexed with bcl2fastq v2.20 (https://support.illumina.com/sequencing/sequencing_software/bcl2fastq-conversion-software.html, accessed on 4 May 2022), and an initial assessment was performed with FastQC v0.11.8. (https://www.bioinformatics.babraham.ac.uk/projects/fastqc/, accessed on 4 May 2022) Salmon v1.5.0 (https://combine-lab.github.io/salmon/, accessed on 4 May 2022) [68] was used to quantify the abundance (TPM, transcripts per million) of expression at the transcript level, using the GRCh37 as the reference. In order to make an assessment of the mapping quality, STAR v2.7.9a (https://github.com/alexdobin/STAR, accessed on 4 May 2022) [69] was used to align reads to reference, Qualimap v2.2.1 (http://qualimap.conesalab.org/, accessed on 4 May 2022) [70] was used to compute quality metrics and MultiQC v1.10.1 (https://multiqc.info/, accessed on 4 May 2022) [71] to aggregate the results, QC metrics and sample data. Finally, differential gene expression analysis was computed with DESeq2 v.1.32.0 (https://bioconductor.org/packages/release/bioc/html/DESeq2.html, accessed on 4 May 2022) [72] package for R. All computations were performed in the TeideHPC supercomputing infrastructure (http://teidehpc.iter.es/en, accessed on 4 May 2022).

### 4.8. Cell Fractioning

For the cell fractioning, two buffer extractions were used, one as Cytosolic Extraction Buffer (CEB) (HEPES 10 mM; KCl 60 mM; EDTA 1 mM; NP40 1%) and another one as Nuclear Extraction Buffer (NEB) (HEPES 20 mM; KCl 500 mM; MgCl2 1.5 mM; EDTA 0.2 mM, glycerol 25% *v*/*v*). First, 2 × 10^6^ CEM.NKR-CCR5 cells or 1 × 10^6^ HeLa P5 cells were washed with 100 μL PBS and centrifuged at 1500 rpm × 5 min. Then, the pellet was resuspended (by gentle pipetting) in CEB, usually about 30–50 μL (5× times the volume of pellet), at 4 °C for 20 min. Centrifugation was performed again at 3000 rpm (720× *g*) for 10 min at 4 °C, and the supernatant was collected as the cytosolic fraction. The pellet (nuclear fraction) was washed at least twice with CEB to eliminate cytoplasmic debris (centrifugation between washes was for 5 min at 4 °C at 3000 rpm), then resuspended in 60 μL NEB (taking into account the nucleus-cytoplasm ratio) and incubated at 4 °C for 30 min, vortexing once every 10 min. At last, the pellet was centrifuged 15 min at 4 °C, 13,000 rpm (10,000× *g*) and the supernatant was collected as the nuclear fraction. For Western-blot analysis, the α-tubulin was used as the cytoplasmic marker and for control of total protein load, and Lamin B1 as the marker for the nuclear fraction.

### 4.9. Western Blotting

Protein expression was determined by SDS-PAGE and Western blotting in cell lysates. Plasmids or siRNA oligos were nucleofected into CEM.NKR-CCR5 cells with Amaxa kits C (1 μg into 2 × 10^6^ cells/mL) or transfected into HeLa P5 cells with polyethylenimine (PEI). PEI with an average molecular mass of 25 kDa (PEI25k) (Polyscience, Warrington, PA, USA) was gently vortexed, incubated for 20–30 min at room temperature (RT) and then added to cells in culture. Briefly, 24 h after nucleofection in the case of overexpression of constructs and 48 h after nucleofection in the case of siRNA oligos, cells were lysated in lysis buffer (1% Triton-X100, 50 mM Tris-HCl pH 7.5, 150 mM NaCl, 0.5% sodium deoxycholate, and protease inhibitor (Roche Diagnostics, GmbH, Mannheim, Germany), for 30 min and sonicated for 30 s at 4 °C. Equivalent amounts of protein (30–40 μg), determined using the bicinchoninic acid (BCA) method (Millipore Corporation, Burlington, MA, USA), were resuspended and treated by Laemmli buffer and then were separated in 10% SDS-PAGE and electroblotted onto 0.45 μm polyvinylidene difluoride membranes (PVDF; Millipore, Burlington, MA, USA) using Trans-Blot Turbo (Bio-Rad, Hercules, CA, USA). Membranes were blocked 5% non-fat dry milk in TBST (100 mM Tris, 0.9% NaCl, pH 7.5, 0.1% Tween 200) for 30 min and then incubated with specific antibodies. Proteins were detected by luminescence using the ECL System (Bio-Rad, Hercules, CA, USA), and analyzed using a ChemiDoc MP device and Image LabTM Software, Version 5.2 (Bio-Rad, Hercules, CA, USA).

### 4.10. Immunofluorescence and Confocal Microscopy Analysis

Control or transfected lymphocytes (with Flag-wt-TDP-43 or Flag-NLS-mut-TDP-43 plasmid (1 μg) using nucleofection kit (Lonza, Verviers, Belgium)) were placed on coverslips (2 × 10^6^ cells in sterile glass coverslips-Ø 12 mm) to be immunolabelled and analyzed by confocal microscopy 48 h post transfection. Then, cells were washed three times with PBS and fixed for 20 min in 2% paraformaldehyde with 1% sucrose in PBS. Next, cells were washed three times with PBS and permeabilized with 0.1% Triton X-100 in PBS. These cells were then washed with PBS and immunostained for 1 h at room temperature (RT) by Alexa 568-labelled goat anti-mouse against Flag-TDP-43 constructs (wt and NLS-mut) previously incubated with a specific mouse mAb (anti-TDP-43 or anti-Flag) and diluted in PBS with 0.1% BSA. Secondary Alexa Fluor 568-goat anti-mouse was used to label TDP-43 (endogenous or Flag-constructs previously bound to mouse anti-TDP-43 mAb), and Alexa Fluor 488-goat anti-rabbit was used to label endogenous HDAC6 (previously bound to rabbit anti-HDAC6 Ab). Dapi was used to stain the nucleus of cells. Then, coverslips carrying these cells were mounted in Mowiol-antifade (Dako, Glostrup, Denmark) and imaged in xy mid-sections in a FluoView FV1000 confocal microscope through a 1.35 numerical aperture (NA) objective (60×) (Olympus, Center Valley, PA, USA) for high-resolution imaging of fixed cells. The final images and molecule codistributions were analyzed with MetaMorph software (Universal Imaging, Downingtown, PA, USA) as previously reported [45,65,66].

### 4.11. Flow Cytometry Analysis

Overexpression effects of the constructs Flag-wt-TDP-43 or Flag-NLS-mut-TDP-43 or the siRNA-TDP-43 on CD4, CCR5 and CXCR4 cell-surface expression in permissive CEM.NKR-CCR5 cells or transfected into HeLa P5 cells, were studied by flow cytometry analysis. Briefly, 24 h nucleofected/transfected cells with fluorescent constructs were incubated in ice-cold (+4 °C) PBS buffer, with an anti-CD4/CCR5/CXCR4 antibody coupled to PE. Labelling of cell-surface receptors was performed by staining with a PE-conjugated IgG isotype. Cells were then washed with ice-cold PBS, fixed in PBS with 1% paraformaldehyde and analyzed by flow cytometry (BD Accuri™ C6 Plus Flow Cytometer, BD Biosciences, San Jose, CA, USA), measuring cell-surface CD4, CCR5 and CXCR4 receptor labelling as described elsewhere [24,26,28,45,65]. Basal cell fluorescence intensity for receptors labelling was determined by staining cells with a PE-conjugated IgG isotype control in cells overexpressing free pCDNA3.1(−) empty vector.

### 4.12. HIV-1 Env Mediated Cell-to-Cell Fusion Assay

A β-Galactosidase cell fusion assay was performed as previously described [45]. Briefly, HeLa-243 or HeLa-ADA cells were mixed with HeLa-P5 cells or control, or previously transfected with TDP-43 constructs or siRNA-TDP-43 oligos, in 96-well plates in a 1:1 ratio (20,000 total cells). These cocultures were kept at fusion for 16 h at 37 °C. The fused cells were washed with Hanks’ balanced salt solution and lysed, and the enzymatic activity was evaluated by chemiluminescence (β-Gal reporter gene assay, Roche Diagnostics, GmbH, Mannheim, Germany). Anti-CD4 neutralizing mAb L3T4 (5 μg/mL) was used as a control for the blockade of cell fusion by preincubating HeLa-P5 cells with this mAb for 30 min at 37 °C before co-culturing with Env+-HeLa cells (243 or ADA).

### 4.13. Production of Viral Particles with Luciferase-Reporter Pseudoviruses

Replication-deficient luciferase-HIV-1 viral particles (luciferase-reporter pseudoviruses) were obtained as described [24,25,26,29,45], using the luciferase-expressing reporter virus HIV/∆*nef*/∆*env*/*luc+* (pNL4-3.Luc.R-E-provirus bearing the luciferase gene inserted into the *nef* ORF and not expressing *env*; catalog number 6070013, NIH AIDS Research and Reference Reagent Program), and the described Env expression plasmids from the different HIV-1 individuals or the reference Env plasmids studied. Thus, HIV-1 viral particles were produced in 12-wells plates by co-transfecting HEK-293T packaging cells (70% confluence) with pNL4-3.Luc.R-E- (1 μg) and R5-tropic (BaL.01), or the X4-tropic HXB2 or a primary Env-glycoprotein vector (1 μg). Viral plasmids were transduced in HEK-293T cells using X-tremeGENE HP DNA transfection reagent (Roche Diagnostics, GmbH, Mannheim, Germany). After the addition of X-tremeGENE HP to the viral plasmids, the solution was mixed in 100 μL of DMEM without serum or antibiotics, and incubated for 20 min at RT prior to adding it to HEK-293T cells. The cells were cultured for 48 h to allow viral production. After this, viral particles were harvested. Viral stocks were normalized by p24-Gag content as measured with an Enzyme Linked Immunosorbent Assay test (GenscreenTM HIV-1 Ag Assay; Bio-Rad, Marnes-la-Coquette, France). Virions were used to infect CEM.NKR-CCR5 cells after ELISA-p24 quantification and normalization. As a control for efficiency of viral production, a CD4 independent viral entry and infection assay was performed in parallel by co-transducing the pNL4-3.Luc.R-E- vector (1 μg) with the pHEF-VSV-G vector (1 μg; National Institutes of Health-AIDS Reagent Program (https://www.hivreagentprogram.org/, 4 May 2022), thereby generating non-replicative viral particles that fuse with cells in a VSV-G-dependent and CD4-independent manner [28].

### 4.14. Luciferase Viral Entry and Infection Assay

CEM.NKR-CCR5 cells (9 × 10^5^ cells in 24-well plates with 20 μg/mL of polybrene) were infected with 200 ng of p24 of luciferase-reporter pseudoviruses, bearing R5-tropic BaL (BaL.01-*env* plasmid, catalog number 11445, NIH AIDS Research and Reference Reagent Program), X4-tropic HXB2 (*env*, catalogue number 5040154, NIH AIDS Research and Reference Reagent Program) or primary Envs in 1 mL total volume with RPMI 1640 for 2 h (by centrifugation at 1200× *g* at 25 °C) and subsequent incubation for 4 h at 37 °C, as previously described [24,25,26,28,29,45,65]. Unbound viruses were then removed by washing the infected cells. After 24 h of infection, luciferase activity was measured using a luciferase assay kit (Biotium, Hayward, CA, USA) with a microplate reader (VictorTM X5; PerkinElmer, Waltham, MA, USA). Anti-CD4 neutralizing mAb L3T4 (5 μg/mL) was used as a control for the blockade of HIV-1 infection by preincubating permissive CEM.NKR-CCR5 cells with this mAb for 30 min at 37 °C before adding viral input.

### 4.15. Statistical Analysis

Statistical analyses were performed using GraphPad Prism, version 6.0b (GraphPad Software, San Diego, CA, USA). Significance when comparing groups was determined with a 2-tailed nonparametric Mann–Whitney U-test.

## Figures and Tables

**Figure 1 ijms-23-06180-f001:**
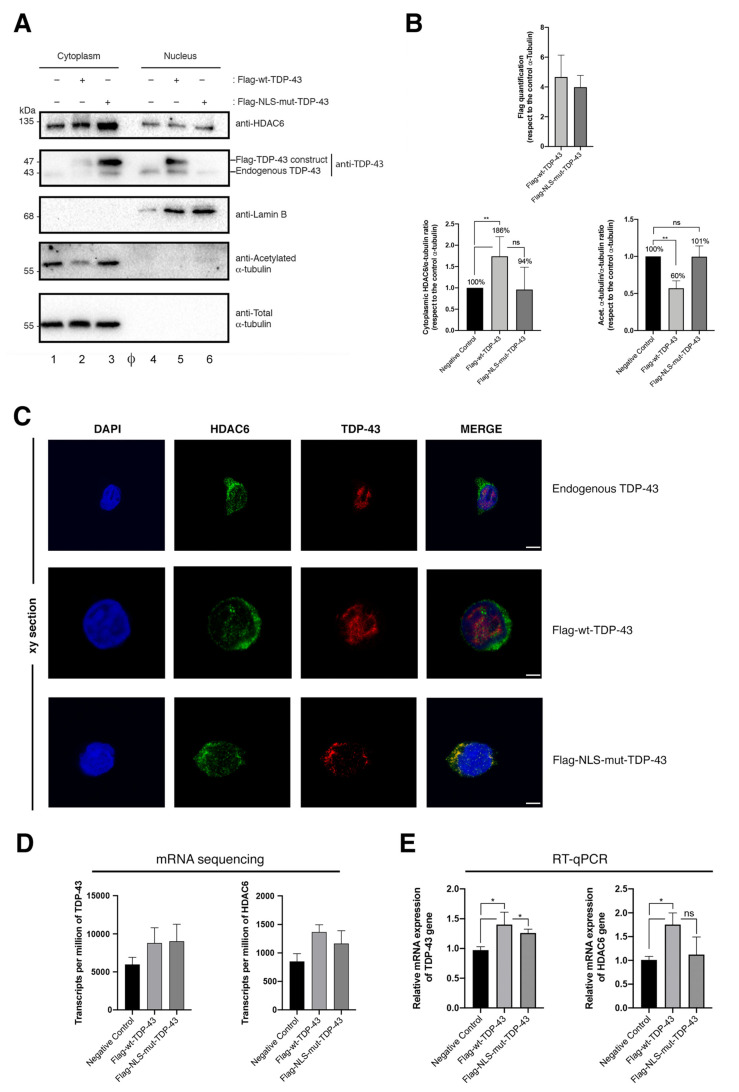
Overexpression of TDP-43 regulates HDAC6 mRNA and protein stability and favors its tubulin-deacetylation function. (**A**) Quantitative Western-blot analysis of cell fractioning from permissive CEM.NKR-CCR5 cells overexpressing Flag-wt-TDP-43 (1 μg cDNA) or Flag-NLS-mut-TDP-43 construct (1 μg cDNA), and their effect on endogenous HDAC6 and α-tubulin acetylation. Lamin B1 is the protein control for the nuclear fraction, whereas α-tubulin for the cytoplasm fraction and total load protein. High acetylated α-tubulin levels are a read-out for MT stabilization after HDAC6-deacetylase activity decreases. A representative experiment of three is shown (see associated data in Appendix A associated information). HDAC6/α-tubulin, acetylated α-tubulin/α-tubulin and Flag/α-tubulin intensity band ratios are shown, where high acetylated α-tubulin levels are a read-out for MT stabilization after HDAC6-deacetylase activity decreases. (**B**) Histograms quantify the intensities of Western-blot bands from panel (**A**) experiment, representing the number of TDP-43 constructs (nuclear Flag-wt-TDP-43 (track 5) by total α-tubulin load control (track 2), and cytoplasmic Flag-NLS-mut-TDP-43 (track 3) by total α-tubulin load control (track 3)), cytoplasmic HDAC6 and acetylated α-tubulin (both from cytoplasmic tracks 1 to 3) by total α-tubulin load respective controls (track 1 to 3). Thus, data are normalized by total α-tubulin load control per each experimental condition (Western-blot pair tracks 1 and 4; 2 and 5, 3 and 6 are from the same experiments). Φ indicates empty track. Data are mean ± standard error of the mean (S.E.M.) of three independent experiments (see associated data in Appendix A associated information). When indicated, ** *p* < 0.05 are *p* values for Student’s *t*-test. ns stands for non-significant. a.u., arbitrary light units. (**C**) Fluorescence confocal microscopy analysis of CEM.NKR-CCR5 cells nucleofected with both TDP-43 constructs, in order to analyze the distribution of the nucleofected proteins. At 48 h post nucleofection, fixed cells were imaged in xy mid-sections in a confocal microscope (Leica TCS SP5; Leica Microsystems, Wetzlar, Germany) in a 1.35 NA objective (60×). Endogenous TDP-43 is located in the nucleus of the cell. Flag-wt-TDP-43 is mainly distributed to the nucleus with some cytoplasmic distribution. TDP-43 does not co-distribute with HDAC6. However, the Flag-NLS-mut-TDP-43 construct is excluded from the nucleus, presenting cytoplasmic localization and abnormal co-distribution with HDAC6. Endogenous HDAC6 is mainly located at the cytoplasm and also presents nuclear distribution. The nucleus is displayed by DAPI labelling. A series of merge images are shown for any experimental condition. A representative experiment of three is shown. Bar, 5 μm (**D**,**E**) Relative mRNA quantification in transcripts per million by sequencing (2 repeats) (**D**) or by RT-qPCR (5 repeats) (**E**) of TDP-43 and HDAC6 genes is represented in histograms under overexpression of TDP-43 constructs. When indicated, * *p* < 0.01 is the *p* value for Student’s *t*-test.

**Figure 2 ijms-23-06180-f002:**
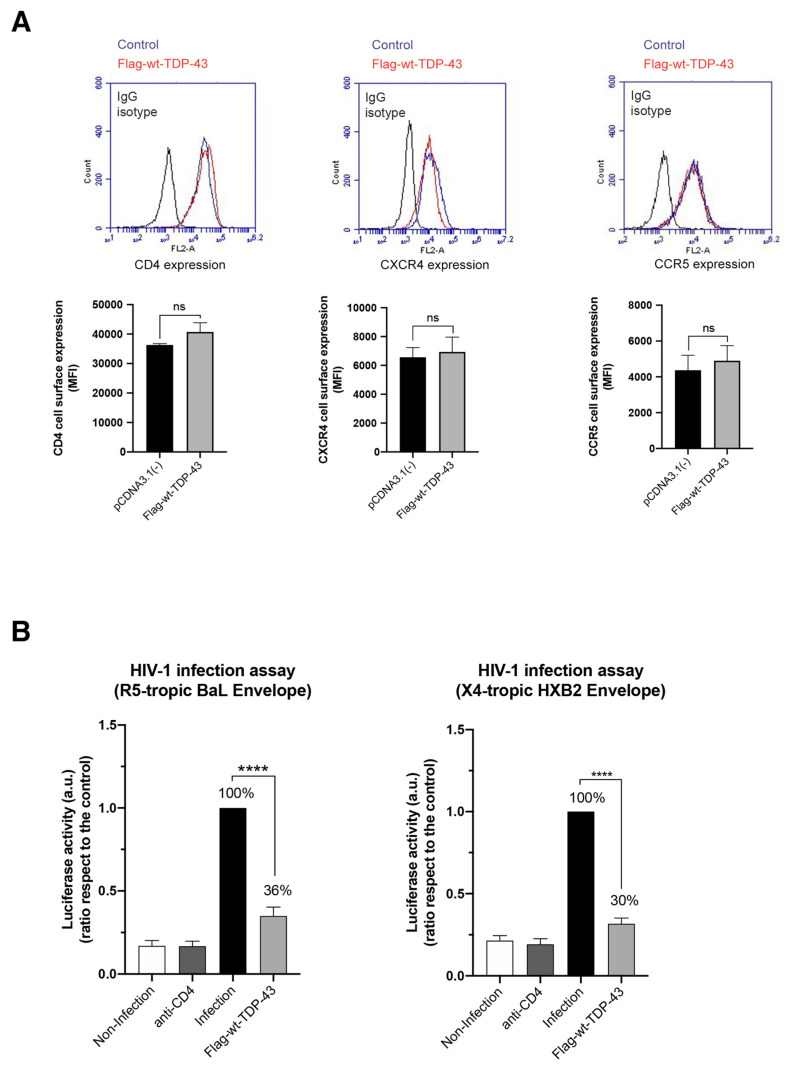
Effect of TDP-43 overexpression on cell-surface expression of HIV-1 main receptor and co-receptors and on viral infection. (**A**) Flow cytometry analysis of CD4, CXCR4 and CCR5 cell-surface expression in CEM.NKR-CCR5+/CD4+ T cells overexpressing the Flag-wt-TDP-43 construct. pcDNA3.1(−) represents control cells nucleofected with the pcDNA3.1 plasmid (1 μg cDNA of each construct). Data are mean ± S.E.M. of three independent experiments carried out in triplicate. Top, representative flow chart with the histograms below where the replicates are shown, indicating CD4, CXCR4 and CCR5 expression, under any experimental condition. Ig isotype control is shown. When indicated ns stands for non-significant. (**B**) Effect of wt-TDP-43 overexpression in permissive CEM.NKR-CCR5 cells on HIV-1 infection. Flag-wt-TDP-43 treated or control cells (nucleofected with the pcDNA3.1 plasmid; 100% infection) were infected with synchronous viral inputs with pNL4-3.Luc.R-E- pseudovirus bearing either R5-tropic BaL or X4-tropic HBX2 viral Env. Nonproductive infection values were obtained with a neutralizing anti-CD4 mAb (5 μg/mL) under the same experimental conditions. Control non-infected cells are shown. Data are mean ± S.E.M. of six independent experiments. When indicated, **** *p* < 0.0001 is the *p* value for Student’s *t*-test; a.u. arbitrary light units.

**Figure 3 ijms-23-06180-f003:**
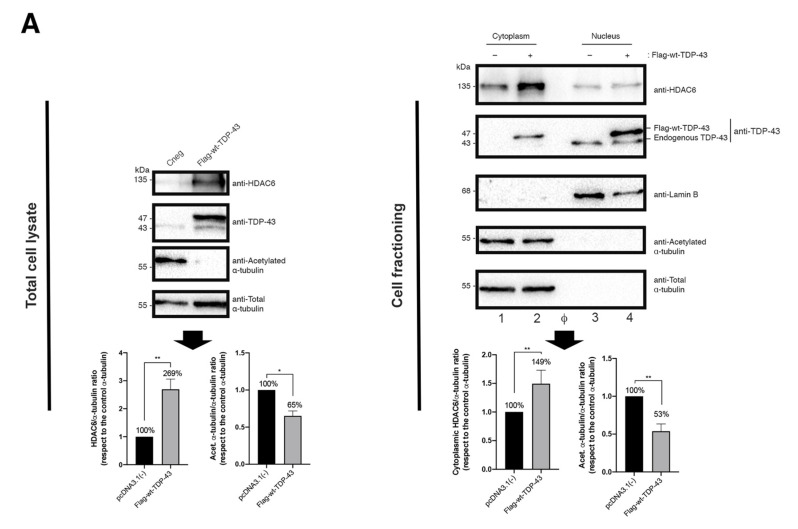
HIV-1 Env fusogenic activity is impaired in cells overexpressing TDP-43. (**A**) Left, total cell lysate, quantitative Western-blot analysis of HDAC6, TDP-43 and acetylated α-tubulin proteins in HeLa P5 cells overexpressing the Flag-wt-TDP-43 construct (1 μg of cDNA) compared to control cells (pcDNA3.1(−) transfected with the pcDNA3.1 plasmid (1 μg)). A representative experiment of three is shown (see associated data in Appendix A associated information). Below, histograms quantify the amounts of HDAC6 and acetylated α−tubulin proteins, normalized by total α-tubulin, in HeLa P5 total cell lysates. Data are expressed as mean ± S.E.M. of three independent experiments. When indicated, * *p* < 0.05 and ** *p* < 0.01 are *p* values for Student’s *t*-test. Right, cell fractioning, quantitative Western-blot analysis of cell fractioning of permissive HeLa P5 cells overexpressing Flag-wt-TDP-43 (1 μg cDNA) and its effect on endogenous HDAC6 and α-tubulin acetylation. Lamin B1 is the protein control for the nuclear fraction, whereas α-tubulin is the control for the cytoplasm fraction and total protein load. The decrease in acetylated α-tubulin levels is a read-out for the increase of the tubulin-deacetylase HDAC6 enzyme. A representative experiment of three is shown (see associated data in Appendix A associated information). Below, histograms quantify the intensities of Western-blot bands from HeLa P5 cell fractioning, representing the amounts of cytoplasmic HDAC6 and acetylated α-tubulin (both from cytoplasmic tracks 1 and 2) in control and Flag-wt-TDP-43 expressing cells, and normalized by total α-tubulin control load (Western-blot tracks 1 and 2). Tracks 3 and 4 are nuclear fractions for control and Flag-wt-TDP-43 expressing cells, respectively. Φ indicates empty track. Data are mean ± standard error of the mean (S.E.M.) of three independent experiments (see associated data in Appendix A associated information). When indicated, ** *p* < 0.05 is the *p* value for Student’s *t*-test. (**B**) Flow cytometry analysis of CD4, CXCR4 and CCR5 cell-surface expression in permissive HeLa P5 cells overexpressing the Flag-wt-TDP-43 construct compared to control cells (transfected with pcDNA.3.1). Data are mean ± S.E.M. of three independent experiments carried out in triplicate. Top, representative flow chart with the histograms below where the replicates are shown, indicating CD4, CXCR4 and CCR5 expression, under any experimental condition. Ig isotype control is shown. (**C**) Relative mRNA quantification by RT-qPCR (3 repeats) of TDP-43 and HDAC6 genes is represented in histograms under overexpression of Flag-wt-TDP-43 construct. When indicated, *** *p* < 0.001 and ** *p* < 0.05 are *p* values for Student’s *t*-test; n.s. stands for non-significative. (**D**) Effect of overexpression of TDP-43 (Flag-wt-TDP-43 construct) in HeLa P5 cells on HIV-1 Env-mediated cell-to-cell fusion activity, measured by β-Galactosidase production and referred to the cell-to-cell fusion control condition (100% of membrane fusion in HeLa P5 cells transfected with pcDNA3.1). HeLa P5 cells alone were used as a control of non-fused cells. Treated HeLa P5 cells were fused with HeLa cells expressing X4-tropic (243) or R5-tropic (ADA) Env, in the presence or the absence of a neutralizing anti-CD4 mAb (5 μg/mL). Data are mean ± S.E.M. of four independent experiments carried out in triplicate. When indicated, *** *p* < 0.001 and **** *p* < 0.0001 are *p* values for Student’s *t*-test. Moreover, the biochemical analysis of protein expression and localization in cellular fractions indicates that endogenous HDAC6 predominantly distributed at the cytoplasm, being also located at the nucleus, whereas endogenous TDP-43 mainly localizes at the nucleus (Figure 3A, *cell fractioning, quantified in the associated histograms below*). We observed that transiently overexpressed Flag-wt-TDP-43 construct mainly localizes in the isolated nuclear fraction of cells (Figure 3A, *cell fractioning, quantified in the associated histograms below*). Under this experimental condition, we observed an increase in the amount of HDAC6 enzyme with a concomitant decrease in acetylated MTs (Figure 3A, *cell fractioning, HDAC6 and acetylated*
*α-tubulin in the associated cytoplasmic fractions, quantified in the associated histograms below*).

**Figure 4 ijms-23-06180-f004:**
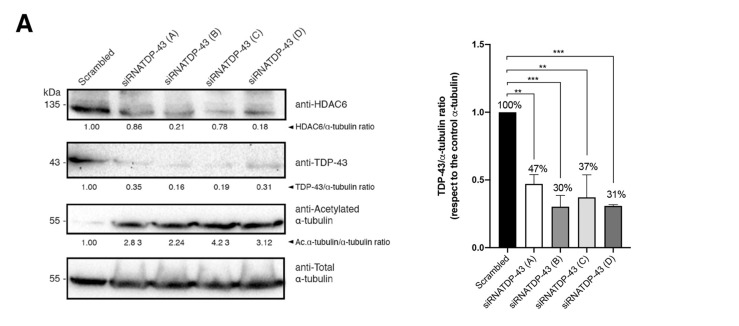
TDP-43 mRNA silencing downregulates HDAC6 mRNA and protein levels. (**A**) Left, Quantitative Western-blot analysis of endogenous TDP-43 knockdown in permissive CEM.NK-CCR5 cells nucleofected with each of the four siRNA-TDP-43 oligos (**A**–**D**) used (1 μM of each oligo). Cells nucleofected with scrambled oligos were used as a control. HDAC6/α-tubulin, TDP-43/α-tubulin and acetylated/α-tubulin/α-tubulin intensity Western-blot band rations are shown. α-tubulin bands represent the control for the total amount of protein. One representative experiment of three is shown (see associated data in Appendix A associated information). Right, histograms show the quantification of the amount of the TDP-43 protein detected, in respect of the α-tubulin protein, after specific siRNA (A to D oligos) silencing and compared to the scrambled condition. Data are mean ± S.E.M. of three independent experiments (see associated data in Appendix A associated information). When indicated, ** *p* < 0.05, and *** *p* < 0.001 are *p* values for Student’s *t*-test. (**B**) Histograms show the quantification of the amount of HDAC6 and acetylated α-tubulin protein detected, with respect to the α-tubulin protein, after specific siRNA (A to D oligos) silencing and compared to the scrambled condition. Data are mean ± S.E.M. of three independent experiments (*see associated data in*
Appendix A
*associated information*). When indicated, * *p* < 0.01, ** *p* < 0.05 and *** *p* < 0.001 are *p* values for Student’s *t*-test. (**C**,**D**) Relative mRNA quantification in transcripts per million by sequencing (1 repeat) (**C**) or by RT-qPCR (3 repeats) (**D**) of TDP-43 and HDAC6 genes is represented in histograms after oligo interference directed to TDP-43 mRNA by either individual siRNA (**A**–**D**) oligo or combining siRNA B + C oligos, and compared to the scrambled condition. Data are normalized to the expression of total α-tubulin. When indicated, * *p* < 0.01 is the *p* value for Student’s *t*-test; n.s. stands for non-significative.

**Figure 5 ijms-23-06180-f005:**
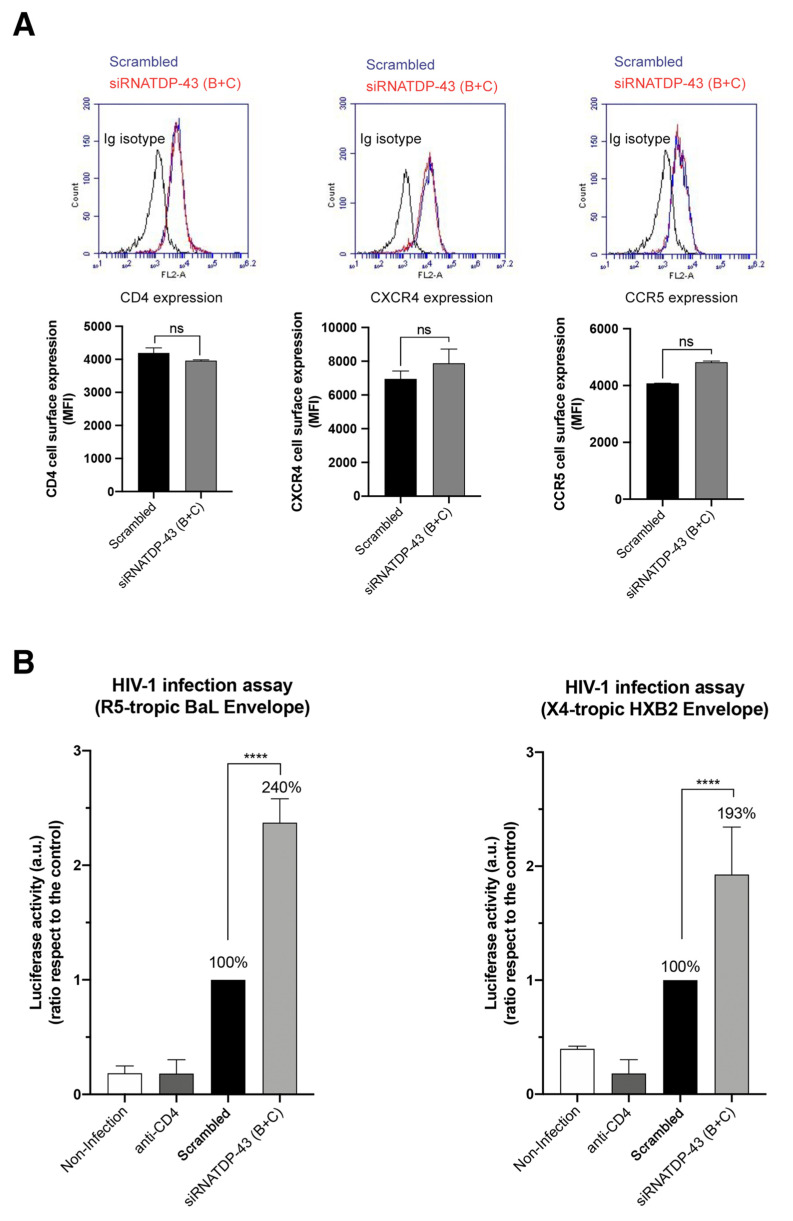
Effect of TDP-43 silencing on cell-surface expression of HIV-1 receptors and on viral infection. (**A**) Flow cytometry analysis of CD4, CXCR4 and CCR5 cell-surface expression in CEM.NKR-CCR5 cells where endogenous TDP-43 were silenced by using specific siRNA oligos (B + C) and compared to scrambled control condition (1 μM of total mix oligos and of scrambled oligo). Data are mean ± S.E.M. of four independent experiments carried out in triplicate. Top, representative flow chart with the histograms below where the replicates are shown, indicating CD4, CXCR4 and CCR5 expression, under any experimental condition. Ig isotype control is shown. When indicated, n.s. stands for non-significative, value for Student’s *t*-test. (**B**) Effect of TDP-43 silencing in permissive CEM.NKR-CCR5 cells on HIV-1 infection. siRNA-TDP-43 (B + C)-treated or scrambled control cells were infected with synchronous viral inputs with pNL4-3.Luc.R-E- pseudovirus bearing either R5-tropic BaL or X4-tropic HBX2 viral Env. Nonproductive infection values were obtained with a neutralizing anti-CD4 mAb (5 μg/mL) under the same experimental condition. Control non-infected cells are shown. Data are mean ± S.E.M. of six independent experiments. When indicated, **** *p* < 0.0001 is the *p* value for Student’s *t*-test.

**Figure 6 ijms-23-06180-f006:**
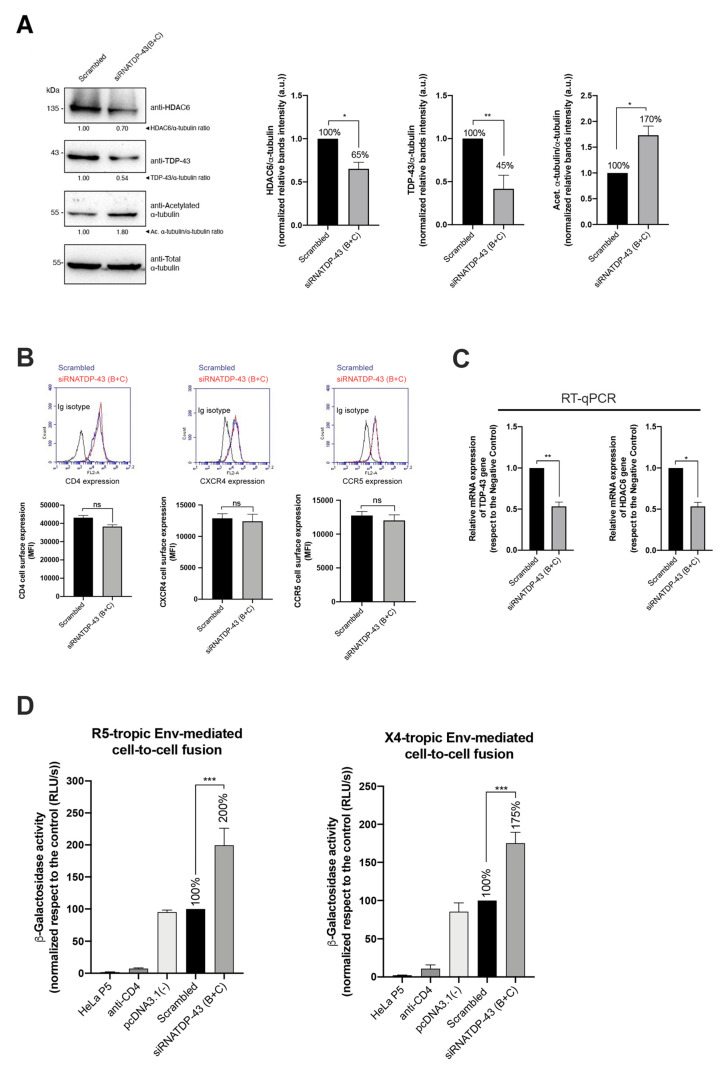
HIV-1 Env fusogenic activity is favored by the interference of the TDP-43 mRNA. (**A**) Left, quantitative Western-blot analysis of endogenous TDP-43, HDAC6 and acetylated α-tubulin proteins in HeLa P5 cells after silencing TDP-43 with specific siRNA oligos (B + C) (1 μM of total mix oligos) and compared to scrambled control condition. A representative experiment of the three is shown (see associated data in Appendix A associated information). Right, histograms quantify the amounts of HDAC6, TDP-43 and acetylated α-tubulin proteins, normalized by total α-tubulin, in HeLa P5 cell lysates from (A) experiments. Data are mean ± S.E.M. of three independent experiments (see associated data in Appendix A associated information). When indicated, * *p* < 0.05 and *** *p* < 0.01 are *p* values for Student’s *t*-test. (**B**) Flow cytometry analysis of CD4, CXCR4 and CCR5 cell-surface expression in permissive HeLa P5 cells where endogenous TDP-43 were silenced by using specific siRNA oligos (B + C) and compared to scrambled control condition. Data are mean ± S.E.M. of three independent experiments carried out in triplicate. Top, representative flow chart with the histograms below where the replicates are shown, indicating CD4, CXCR4 and CCR5 expression, under any experimental condition. Ig isotype control is shown. (**C**) Relative mRNA quantification by RT-qPCR (3 repeats) of TDP-43 and HDAC6 genes is represented in histograms under interference of TDP-43 mRNA with silencing oligonucleotide pull (siRNATDP-43 (B + C)). When indicated, * *p* < 0.01 and ** *p* < 0.05 are *p* values for Student’s *t*-test; n.s. stands for non-significative. (**D**) Effect of specific TDP-43 silencing (siRNATDP-43 (B + C) oligos) in HeLa P5 cells on HIV-1 Env-mediated cell-to-cell fusion activity, measured by β-Galactosidase production and compared to the cell-to-cell fusion in scrambled, control condition. Treated HeLa P5 cells were fused with HeLa cells expressing X4-tropic (243) or R5-tropic (ADA) Env, in the presence or the absence of a neutralizing anti-CD4 mAb (5 μg/mL). Cell-to-cell fusion extension with HeLa P5 cells transfected with the pcDNA3.1 plasmid is compared with the scrambled condition. HeLa P5 cells alone were used as a control of non-fused cells. Data are mean ± S.E.M. of four independent experiments carried out in triplicate. When indicated, *** *p* < 0.001 is the *p* value for Student’s *t*-test.

**Figure 7 ijms-23-06180-f007:**
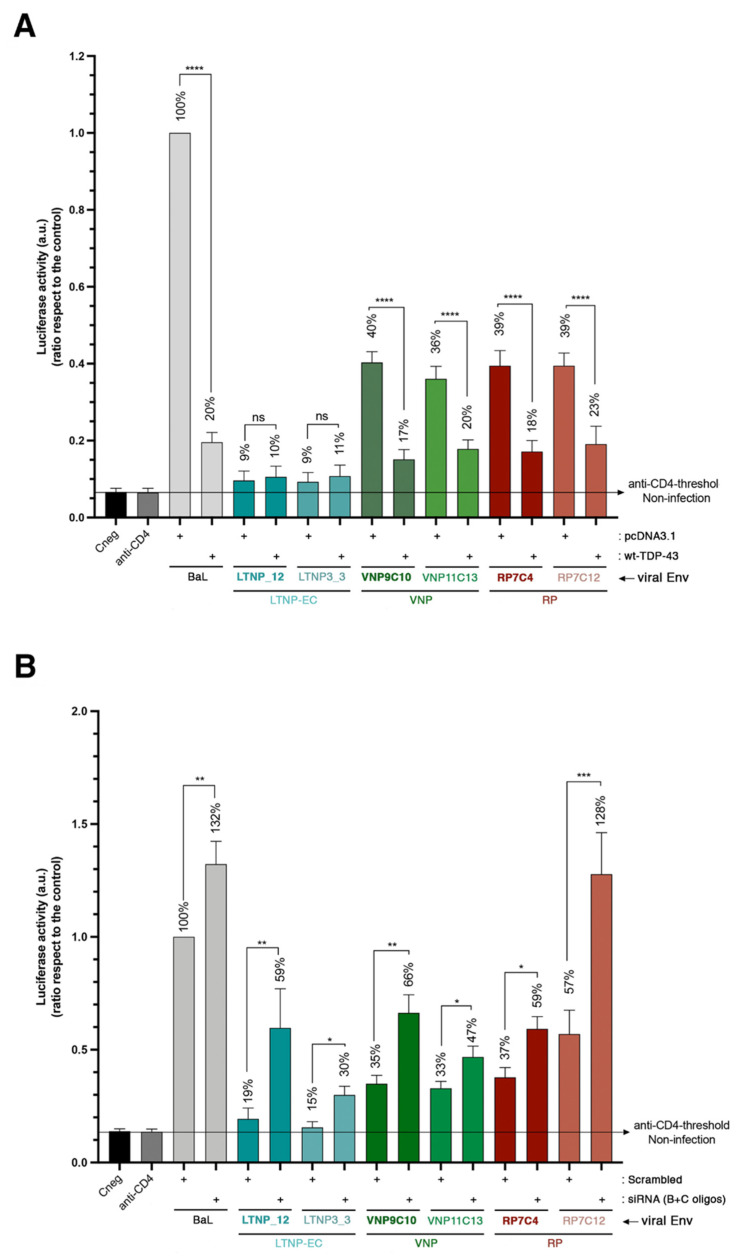
TDP-43 regulates cell HIV-1 infection by envelopes of primary viruses from patients of extreme clinical phenotypes. (**A**) Effect of wt-TDP-43 overexpression (1 μg of cDNA) in permissive CEM.NKR-CCR5 cells on luciferase-based infection assay by using synchronous viral inputs of nonreplicative HIV-1 pseudoviruses bearing R5-tropic primary viral Envs from virus of LTNP-EC (LTNP1_12 and LTNP3_3 Envs), VNP (VNP9C10 and VNP11C13 Envs) and RP (RP7C4 and RP7C12 Envs) patients. These viral infections were compared with those on control cells (nucleofected with pcDNA3.1; 1 μg of cDNA). Nonproductive infection values (baseline) were obtained with a neutralizing anti-CD4 mAb (5 μg/mL) and cells infected with virus bearing the R5-tropic BaL Env are considered 100% of maximum infection, under the same experimental conditions. Control non-infected cells (Cneg) are shown. Data are mean ± S.E.M. of six independent experiments. When indicated, **** *p* < 0.0001 is the *p* value for Student’s *t*-test; n.s. stands for non-significative. (**B**) Effect of specific of TDP-43 silencing, by using siRNA (B + C) oligos (1 μM of total mix, and 1 μM for control scrambled oligo), in permissive CEM.NKR-CCR5 cells on luciferase-based infection assay, infected with synchronous viral inputs of nonreplicative HIV-1 pseudoviruses bearing R5-tropic primary viral Envs from virus of LTNP-EC (LTNP1_12 and LTNP3_3 Envs), VNP (VNP9C10 and VNP11C13 Envs) and RP (RP7C4 and RP7C12 Envs) patients. These viral infections were compared with those on scrambled control cells. Nonproductive infection values (baseline) were obtained with a neutralizing anti-CD4 mAb (5 μg/mL) and cells infected with virus bearing the R5-tropic BaL Env are considered 100% of maximum infection, under the same experimental conditions. Control non-infected cells (Cneg) are shown. Data are mean ± S.E.M. of six independent experiments. When indicated, * *p* < 0.05, ** *p* < 0.01 and *** *p* < 0.001 are *p* values for Student’s *t*-test.

**Figure 8 ijms-23-06180-f008:**
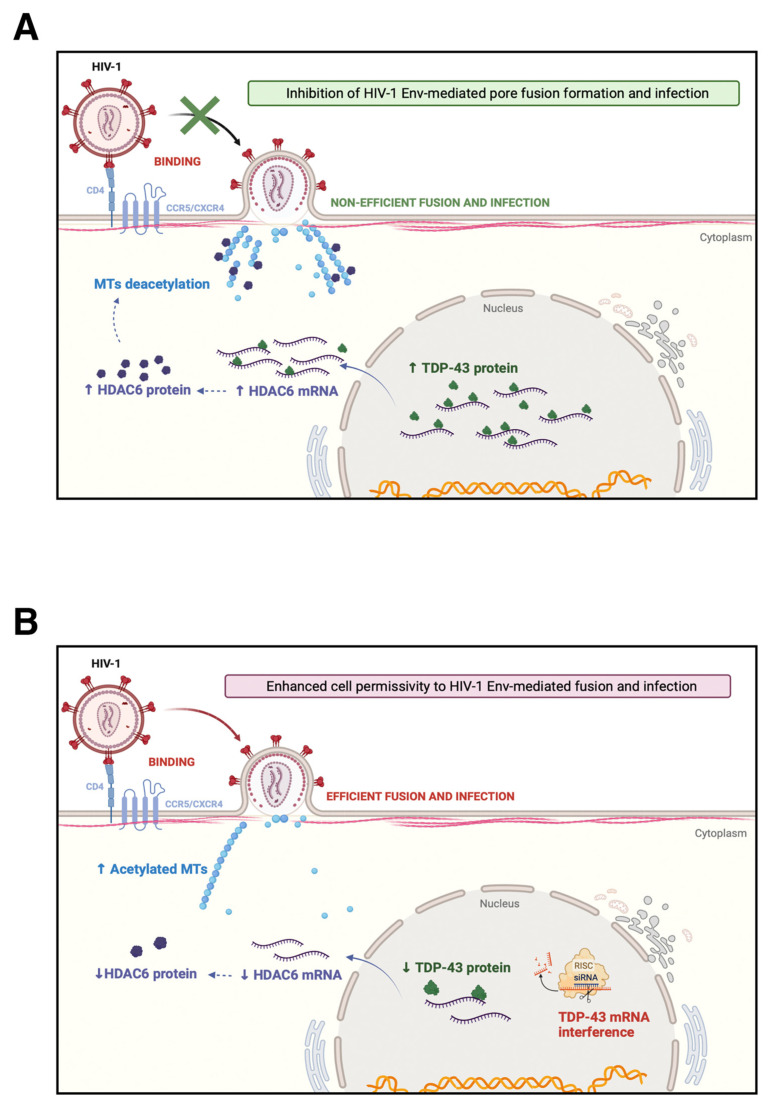
Schematic representation of the regulatory mechanism of CD4+ T-cell permissivity to HIV-1 infection exerted by the TDP-43/HDAC6 axis. (**A**) TDP-43 overexpression increased both mRNA and protein levels of the tubulin-deacetylase HDAC6 enzyme, which led to a decrease in the level of stable acetylated MTs required for efficient HIV-1 Env-mediated pore fusion formation and infection [52,63]. This cellular condition imposes a restriction for HIV-1 infection, independently of the tropism of the viral Env complex. Thus, an increase in the TDP-43/HDAC6 axis markedly reduced the infection activity of viral Envs of virus from VNP and RP patients down to the levels of the inefficient HIV-1 Envs of virus from LTNP-EC individuals. (**B**) Consistently, loss of the endogenous TDP-43 protein by specific siRNA silencing of TDP-43 mRNA results in a significant decrease in both HDAC6 mRNA and the enzyme. Cells present an increase of stable acetylated MTs that enhance cell permissivity to HIV-1 Env-mediated pore fusion formation and infection, independently of the tropism of the viral Env complex. Furthermore, silencing of endogenous TDP-43 significantly favored the infectivity of primary Envs of virus from VNP and RP patients, and importantly increases the infectivity of those from LTNP-ECs. Therefore, TDP-43 shapes cell permissivity to HIV-1 infection, affecting viral Env fusion and infection capacities by altering the HDAC6 levels and associated tubulin-deacetylase anti-HIV-1 activity. Schemes were created with BioRender.com (accessed on 4 May 2022).

## Data Availability

The datasets presented in this study can be found in an online repository. The name of the repository and the accession number can be found below: https://zenodo.org/ (accessed on 4 May 2022), accessions: 10.5281/zenodo.5897655 and 10.5281/zenodo.5901179 (accessed on 4 May 2022).

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
