# Peer review of "Transactive Response DNA-Binding Protein (TARDBP/TDP-43) Regulates Cell Permissivity to HIV-1 Infection by Acting on HDAC6"

_ijms, 2022, doi:10.3390/ijms23116180_

Round 1
Reviewer 1 Report
The submitted manuscript from Cabrera-Rodriguez et al. intends to demonstrate that TDP-43 can contribute to the inhibition of HIV-1 infection, possibly through TDP-43 functional activity on HDAC6 mRNA stabilization thus leading to optimal HDAC6-mediated viral restriction. Authors further showed that the effect of TDP-43 is independent of viral tropism but could influence the infection outcome depending on clinical status and disease progression.
The proposed study is of interest to the field related to HIV-1 infection and reveals a novel pathway linking TDP-43 and HDAC6 in the inhibition of HIV-1 infection of CD4+ T cells. Most experiments are well-conducted, technically sound and results detailed enough to clearly understand experimental outcome. One weakness, at least, is the inadvertent shortcut made by authors when concluding that the obtained viral restriction phenotype would occur solely through HDAC6 mRNA. Although HDAC6 mRNA levels can be indeed a target of TDP-43 it might not be the only one and authors have not modulated HDAC6 mRNA levels in parallel (genetically or pharmacologically) to preclude this. Authors need therefore to tone down this aspect and possibly keep a conditional tense in the discussion part.
The article is nicely presented and figures are of good quality. The graphical scheme of the model is particularly well appreciated. One of the strengths of the study is the use of clinical HIV-1 envelopes for which sequences were subcloned in HIV-derived lentivectors thus offering an original screening of TDP-43-mediated effect on infectivity assays related to disease progression.
The manuscript is quite long with many replicates for all data for which their input is coherent and justified in the flow of the manuscript. Article is well-written but would need some English language correction here and there. There are few minor concerns which need however to be addressed by authors:
- - As per quantifications in Figure 1B, authors need to show the anti-FLAG western-blotting data in Figure 1A thus precluding any potential artefactual detection of ectopic WT and mutant TDP-43 with anti-TDP-43. This is particularly important when considering the high propensity of TDP-43 to be phosphorylated for which migration profile could be mistaken with the tagged versions of TDP-43.
- - The effects obtained with the TDP-43 mutant are puzzling. Have the authors an explanation for why the HDAC6 in cells overexpressing a NLS-mutant TDP-43 is seemingly not able anymore to deacetylate alpha-tubulin? Also, quantification results for HDAC6/alpha-tubulin ratio presented in 1B (lower graphs) are weird and not seemingly representing WB data from 1A particularly when considering NLS-mutant TDP-43 conditions. Although WB is not quantitative and human eye is definitely not adapted for those gray nuances, can the authors please verify their quantification data. The same holds true with RT-qPCR data for HDAC6 mRNA expression (Fig1E) which do not seem coherent with the increased HDAC6 levels by WB in Fig1A upon NLSmutant-TDP43 expression!!?
- - Because of the apparent discrepancy and incoherent data obtained with TDP-43 NLS-mutant authors might envisage to remove this from the manuscript as it doesn’t neither help the understanding nor influence the main conclusion of their study. This should be anyway the case in the flow cytometry data from Figure 2 because the TDP-43 mutant is not used further.
- - Figure 2B is lacking and Figure 2C is therefore miscounted
- - Are graphs in Figure 3C representing folds compared to NegCtrl? If so, please specify in axis title.
- - Please replace the term “permissibility” by permissivity throughout all manuscript.
- - Please also, specify HIV-1 throughout all the manuscript
Other Minor comments
- - The reader would have appreciated authors to describe individual panels of Figure 6 instead of treating this Figure as a whole.
- - Given authors’ past reports on HDAC6-mediated restriction of HIV-1 infection, It would have been appreciated to discuss some of the current described aspects in relation to the autophagy pathway. How and where the authors would place the HDAC6-mediated effects
- Lines 57-59: weird and confusing sentence. Please modify.
- Lines 71-75: This sentence is too long which renders it confusing
- Line 142: please correct “CEMM…”
- Line 161: please replace “microscopy” by microscope
- Line 239: please correct “tubulind”
- Figure 3D: please correct R5-topic and X4-topic
- Line 405: please modify “increase” by increases
- Sentences in Line 429 to 432 need to be revised
- Starting sentence in Line 822 needs to be corrected
- Line 824: please modify “cleavages” by cleaves
Author Response
Point-by-point reply to the Reviewers
Reviewer#1:
The submitted manuscript from Cabrera-Rodriguez et al. intends to demonstrate that TDP-43 can contribute to the inhibition of HIV-1 infection, possibly through TDP-43 functional activity on HDAC6 mRNA stabilization thus leading to optimal HDAC6-mediated viral restriction. Authors further showed that the effect of TDP-43 is independent of viral tropism but could influence the infection outcome depending on clinical status and disease progression.
The proposed study is of interest to the field related to HIV-1 infection and reveals a novel pathway linking TDP-43 and HDAC6 in the inhibition of HIV-1 infection of CD4+ T cells. Most experiments are well-conducted, technically sound and results detailed enough to clearly understand experimental outcome.
We thank Reviewer#1 for this summary of our work and the kind comments on our work also highlighting the role of the TDP-43/HDAC6 axis on HIV-1 infection.
One weakness, at least, is the inadvertent shortcut made by authors when concluding that the obtained viral restriction phenotype would occur solely through HDAC6 mRNA. Although HDAC6 mRNA levels can be indeed a target of TDP-43 it might not be the only one and authors have not modulated HDAC6 mRNA levels in parallel (genetically or pharmacologically) to preclude this. Authors need therefore to tone down this aspect and possibly keep a conditional tense in the discussion part.
We agree with this comment made by Reviewer#1. Therefore, we have indicated in greater clarity that although we observed that HDAC6 mRNA levels can be targeted by TDP-43 it might not be the only mRNA targeted that may affect HIV-1 infection.
This idea has been introduced in the Discussion section of the Ms (page 17, lines 809-812, after “(summarized in the schemes of Figure 8)”; text marked in red-yellow), as follows:
“Although it was observed that HDAC6 mRNA is targeted by TDP-43, thereby affecting HDAC6-mediated anti-HIV-1 activity, this might not be the only mRNA associated with proteins that may affect HIV-1 infection targeted by TDP-43.”
The article is nicely presented and figures are of good quality. The graphical scheme of the model is particularly well appreciated. One of the strengths of the study is the use of clinical HIV-1 envelopes for which sequences were subcloned in HIV-derived lentivectors thus offering an original screening of TDP-43-mediated effect on infectivity assays related to disease progression.
The manuscript is quite long with many replicates for all data for which their input is coherent and justified in the flow of the manuscript.
We would like to thank Reviewer #1 for these kind comments about our work.
Article is well-written but would need some English language correction here and there.
The manuscript was English language edited by Patrick Dennis, an experienced English language editor of scientific manuscripts for international publications.
We have acknowledged this fact in the Acknowledgments section of the Ms (page 24, lines 1171-1173; text marked in red-yellow), as follows:
“We would like to thank Patrick Dennis for his support. The manuscript was English language edited by Patrick Dennis, an experienced English language editor of scientific manuscripts for international publications.”
We have highlighted all the main modifications in red-yellow color, in the text of the Ms (ijms-1736305).
There are few minor concerns which need however to be addressed by authors:
- As per quantifications in Figure 1B, authors need to show the anti-FLAG western-blotting data in Figure 1A thus precluding any potential artefactual detection of ectopic WT and mutant TDP-43 with anti-TDP-43. This is particularly important when considering the high propensity of TDP-43 to be phosphorylated for which migration profile could be mistaken with the tagged versions of TDP-43.
We have only shown TDP-43 associated bands revealed by using an anti-TDP-43 Ab, since phosphorylation could also occur in the Flag-tagged TDP-43 constructs and we did not observe bands at a higher relative molecular weight (MW) that would correspond to the phosphorylated forms of the Flag-constructs. Moreover, if the phosphorylation had occurred, we would also have observed endogenous phosphorylated forms even in control conditions, but we did not observe any other TDP-43 associated bands, under this experimental condition. Therefore, higher bands correspond to the Flag-tagged TDP-43 constructs, only observed when Flag constructs have been over-expressed. For example, nucleus Flag NLS-mutant condition (track 6) and nucleus control (track 4) did not display a higher relative MW band as observed in track 5 (Flag-wt TDP-43 construct). Similar results could be observed in cytoplasm associated tracks (1 to 3).
Moreover, fluorescence confocal microscopy analysis of CEM.NKR-CCR5 cells nucleofected with both Flag-tagged TDP-43 constructs (Figure 1C) indicates that Flag-wt-TDP-43 is mainly distributed to the nucleus with some cytoplasmic distribution. Wt-TDP-43 does not co-distribute with HDAC6. However, the Flag-NLS-mut-TDP-43 construct is excluded form the nucleus, presenting cytoplasmic localization and abnormal co-distribution with HDAC6. Therefore, we observed that the TDP-43 constructs and the endogenous TDP-43 presented a cellular localization that is conceivable with a non-inducible phosphorylated status under our experimental conditions.
In this matter, it has been reported that a non-phosphorylatable TDP-43 mutant (S409A/S410A) remains in the nucleus in rats and rat primary neurons models of intercranial hemorrhage (ICH), suggesting a role in this for phosphorylation in TDP-43 cytoplasmic localization (Sun L, et al. TAR DNA binding protein-43 loss of function induced by phosphorylation at S409/410 blocks autophagic flux and participates in secondary brain injury after intracerebral hemorrhage. Front Cell Neurosci. 2018;12:79). Furthermore, following ER stress, the kinase CK1 promotes TDP-43 cytoplasmic accumulation in NSC-34 cells (Hicks DA, et al. Endoplasmic reticulum stress signalling induces casein kinase 1-dependent formation of cytosolic TDP-43 inclusions in motor neuron-like cells. Neurochem Res. 2020;45(6):1354–64). Expression of phosphomimetic mutations S375E and S387E/ S389E/S393E/S395E in TDP-43’s low-complexity domain also significantly increases TDP-43 cytoplasmic localization compared to wild type in HeLa cells (Newell K, et al. Dysregulation of TDP-43 intracellular localization and early onset ALS are associated with a TARDBP S375G variant. Brain Pathol. 2019;29(3):397–413). Therefore, inducible phosphorylation of TDP-43 would affect its nuclear localization, thereby promoting its translocation to the cytoplasm (reviewed in (Eck RJ, Kraemer BC, Liachko NF. Regulation of TDP-43 phosphorylation in aging and disease. Geroscience. 2021 Aug;43(4):1605-1614. doi: 10.1007/s11357-021-00383-5. Epub 2021 May 25. PMID: 34032984)). An event that we did not observed in control conditions as well as in cells over-expressing wt-TDP-43.
These data confirm that we are able to detect the over-expressed Flag-TDP-43 constructs by western-blot and fluorescence confocal microscopy, not corresponding to a phosphorylatable status of the endogenous TDP-43. Therefore, we decided to only show the bands detected by using the anti-TDP-43 Ab to ease the reading of panel A.
- The effects obtained with the TDP-43 mutant are puzzling. Have the authors an explanation for why the HDAC6 in cells overexpressing a NLS-mutant TDP-43 is seemingly not able anymore to deacetylate alpha-tubulin? Also, quantification results for HDAC6/alpha-tubulin ratio presented in 1B (lower graphs) are weird and not seemingly representing WB data from 1A particularly when considering NLS-mutant TDP-43 conditions. Although WB is not quantitative and human eye is definitely not adapted for those gray nuances, can the authors please verify their quantification data.
The same holds true with RT-qPCR data for HDAC6 mRNA expression (Fig1E) which do not seem coherent with the increased HDAC6 levels by WB in Fig1A upon NLSmutant-TDP43 expression!!?
We observed that the ΔNLS TDP-43 mutant abnormally co-localizes with endogenous HDAC6 in some kind of aggregates structures in the cytoplasm of the cell (Figure 1C, Flag-mut-TDP-43 Merge images),. Under this condition, the level of MTs acetylation in the a-tubulin subunit are compared to those observed in the control condition (Figure 1A, cytoplasm tracks 1 and 3, respectively). This fact could be explained by the inability of the sequestered HDAC6 to be associated with MTs in these ΔNLS TDP-43 mutant-mediated aggregates. This fact is not observed with the endogenous TDP-43 or the wt-TDP-43 construct (Figure 1C, endogenous and Flag-wtTDP-43 series of images). Therefore, the sequestered HDAC6 in the ΔNLS TDP-43 mutant-elicited structures may not deacetylate MTs in their a-tubulin subunit. In this regard, it has been reported that the interaction of HDAC6 with SQSTM1/p62 hinders the deacetylase activity of HDAC6 (Yan J, et al. (2013). SQSTM1/p62 interacts with HDAC6 and regulates deacetylase activity. PLoS One. Sep 27;8(9):e76016. doi: 10.1371/journal.pone.0076016. PMID: 24086678).
More work is required to establish the events behind this observation, and we therefore prefer not to develop a discussion at that level. However, and considering this ideas, we have modified the sentence in lines 209-212 (page 5), as follows:
“Perhaps its abnormal cytoplasmic localization in aggregates where HDAC6 co-localizes may alter the HDAC6 tubulin-deacetylase activity, as it has been reported by the interaction of TDP-43 with p62 and HDAC6 at the cytoplasm [15,40-44].”.
Regarding the levels of the HDAC6 enzyme and mRNA observed in cells expressing the ΔNLS TDP-43 mutant, it is conceivable that the sequestered HDAC6 enzyme in the ΔNLS TDP-43 mutant-mediated aggregates will not follow a normal recycling process, thereby observing an increase of the levels in some of the experiment replicates. However, the ΔNLS TDP-43 mutant in the aggregates it forms at the cytoplasm will not alter the HDAC6 mRNA levels compared to control condition (Figure 1E, negative control). This fact could explain the levels of HDAC6 protein and mRNA observed in cells over-expressing the ΔNLS TDP-43 mutant.
The different replicates of the experiment show the values obtained for HDAC6 protein and mRNA in cells over-expressing the ΔNLS TDP-43 mutant, as we have represented in the associated histograms for quantification.
- Because of the apparent discrepancy and incoherent data obtained with TDP-43 NLS-mutant authors might envisage to remove this from the manuscript as it doesn’t neither help the understanding nor influence the main conclusion of their study. This should be anyway the case in the flow cytometry data from Figure 2 because the TDP-43 mutant is not used further.
As suggested by Reviewer#1, we have removed the NLS mutant associated data from Figure 2 (also corrected in the associated figure legend and main text (page 5, Lines 225-228)).
However, we would like to keep the associated results in Figure 1, since this construct was used as a control for wt-TDP-43 expression and cell localization.
We have also adapted the 2.1 (page 2, line 90) and 2.2. (page 4, line 195) subheadings, in the Results section (page 2, line 90), as follows:
“2.1. Characterization of the expression and cellular localization of TDP-43”.
“2.2. TDP-43 stabilizes HDAC6 and diminishes a-tubulin acetylation”.
Figure 2B is lacking and Figure 2C is therefore miscounted
We agree with this observation made by Reviewer#1. We have not presented data for cell-surface distribution of the receptors. Therefore, Figure 2C should not have appeared in the manuscript.
We have modified the sentence (changes in line 225 and line 230, page 5; text marked in red-yellow), as follows:
“We observed that over-expression of wt-TDP-43 did not significantly change the cell-surface expression levels of CD4, CCR5 and CXCR4 molecules (i.e., main receptor and co-receptors for HIV-1 infection, respectively) (Figure 2A). The cell infection using pseudovirus bearing BaL or HXB2 Env was notably diminished in cells over-expressing wt-TDP-43 compared to control, untreated cells (Figure 2B).”
- Are graphs in Figure 3C representing folds compared to NegCtrl? If so, please specify in axis title.
These data are represented respect to the control (NegCtrl) and, as requested by Reviewer#1, we have indicated this fact in the Y-axis title, as follows:
“Relative mRNA expression of TDP-43 gene (respect to the Negative control)”
- Please replace the term “permissibility” by permissivity throughout all manuscript.
We thank Reviewer#1 for this comment and “permissibility” has now been replaced by “permissivity” throughout the manuscript.
- Please also, specify HIV-1 throughout all the manuscript
We thank Reviewer#1 for this comment and HIV-1 has now been specified throughout the manuscript.
Other Minor comments
- The reader would have appreciated authors to describe individual panels of Figure 6 instead of treating this Figure as a whole.
We agree with this observation made by Reviewer#1. However, we aimed to present the TDP-43 and HDAC6 expression levels, in cells treated by specific siRNA-TDP-43 oligos, together with the results of HIV-1 Env-mediated cell-to-cell fusion, under this experimental condition. This avoids the need to explain all the experimental conditions and fusion in two-three different figures.
We hope Reviewer#1 understands that we prefer to maintain the original Figure 6.
- Given authors’ past reports on HDAC6-mediated restriction of HIV-1 infection, It would have been appreciated to discuss some of the current described aspects in relation to the autophagy pathway. How and where the authors would place the HDAC6-mediated effects
We thank Reviewer#1 for this suggestion that we have discussed on page 17 (lines 823-831; text marked in red-yellow), as follows:
“The authors have previously reported the anti-HIV-1 activity of HDAC6 by targeting the Pr55Gag and Vif key viral proteins to the autophagy degradative pathway, inhibiting viral production and virion infectiveness but promoting the stabilization of the anti-HIV-1 restriction factor A3G [25,53]. Furthermore, productive early HIV-1 infection requires the inhibition of autophagy and its related machinery, such as the pro-autophagic and anti-HIV-1 HDAC6 enzyme, whereas non-productive signaling in bystander target cells promotes late autophagy and subsequent cell death [25,26,29,48,54,55]. Therefore, it is plausible that the TDP-43 action on HDAC6 could also control HIV-1 infection and replication by modulating the HDAC6-mediated autophagy anti-HIV-1 functions.”
- Lines 57-59: weird and confusing sentence. Please modify.
We have modified the sentence (page 2, lines 57-59), as follows:
“One of the transcripts that is mainly regulated by TDP-43 is that corresponding to the cytoplasmic enzyme, histone deacetylase 6 (HDAC6)”
- Lines 71-75: This sentence is too long which renders it confusing
We have modified the sentence (page 2, now lines 71-74), as follows:
“It is important to note that HDAC6 is key to regulating HIV-1 infection. HDAC6 affects pore fusion formation by preventing HIV-1-mediated -tubulin acetylation of MTs, and promoting HDAC6/p62 autophagy to clear key viral factors, such as Vif and Pr55Gag thereby stabilizing the HDAC6/APOBEC3G (A3G) restriction factor complex”.
- Line 142: please correct “CEMM...”
We thank Reviewer#1 for this comment. We have corrected this term, as follows:
“CEM.NKR-CCR5 cells”
- Line 161: please replace “microscopy” by microscope
We thank Reviewer#1 for this comment. “microscopy has now been replaced by “microscope” in new line 159.
- Line 239: please correct “tubulind”
We thank Reviewer#1 for this comment. This has now been corrected.
- Figure 3D: please correct R5-topic and X4-topic
We thank Reviewer#1 for this comment. These terms have now been corrected.
- Line 405: please modify “increase” by increases
We thank Reviewer#1 for this comment: “increase” has been replaced by “increases” (new line 410, page 9).
- Sentences in Line 429 to 432 need to be revised
We have modified the sentence (page 9, new lines 434-438), as follows:
“We then used a combination of two oligos, siRNA-TDP-43 B and C, in order to silence TDP-43. Under this experimental condition, we assayed HIV-1 Env-mediated pore fusion formation, entry an infection. We first analyzed how siRNA-TDP-43 silencing affect the ability of the viral Env to promote pore fusion formation, by using the HIV-1 Env-mediated cell-to-cell fusion model.”
- Starting sentence in Line 822 needs to be corrected
We have modified the sentence (page 17, new lines 820-822), as follows:
“The TDP-43/HDAC6 axis could be another factor that is worth exploring as well as the immune responses in EC individuals that lose their status of natural controllers of the infection [29,51,52]”.
- Line 824: please modify “cleavages” by cleaves
We thank Reviewer#1 for this comment. We have now replaced “cleavages” by “cleaves”.
Now, line 835 reads as follows:
“This viral protease cleaves TDP-43 affecting…”

Reviewer 2 Report
The manuscript Transactive response DNA-binding Protein (TARDBP/TDP-43) regulates cell permissibility to HIV-1 infection by acting on HDAC6 brings some interesting data, results are extensively presented, the topic thoroughly researched by many articles the authors refer. I advise English language corrected by the native speaker or English literature professor and for the results presentation to be revised, maybe there is room for some cuts and placing some figures/tables from the original article to supplementary section but other than that I find the manuscript ready for publishing.
Author Response
Point-by-point reply to the Reviewers
Reviewer#2
The manuscript Transactive response DNA-binding Protein (TARDBP/TDP-43) regulates cell permissibility to HIV-1 infection by acting on HDAC6 brings some interesting data, results are extensively presented, the topic thoroughly researched by many articles the authors refer.
I advise English language corrected by the native speaker or English literature professor and for the results presentation to be revised, maybe there is room for some cuts and placing some figures/tables from the original article to supplementary section but other than that I find the manuscript ready for publishing.
We would like to thank Reviewer#2 for these kind comments about our work.
The manuscript was English language edited by Patrick Dennis, an experienced English language editor of scientific manuscripts for international publications.
We have acknowledged this fact in the Acknowledgments section of the Ms (page 24, lines 1171-1173); text marked in red-yellow), as follows:
“We would like to thank Patrick Dennis for his support. The manuscript was English language edited by Patrick Dennis, an experienced English language editor of scientific manuscripts for international publications.”
We have highlighted all the main modifications in red-yellow color, in the text of the Ms (ijms-1736305).
